# FABM-NflexPD 1.0: Assessing an Instantaneous Acclimation Approach for Modelling Phytoplankton Growth

Onur Kerimoglu[1,2], Prima Anugerahanti[3], and Sherwood Lan Smith[3]

[1]Institute for Chemistry and Biology of the Marine Environment, University of Oldenburg, Germany
[2]Helmholtz Center for Coastal Research, Germany
[3]Earth SURFACE Research Center, Research Institute for Global Change, JAMSTEC, Japan

**Correspondence:** Onur Kerimoglu (kerimoglu.o@gmail.com)

**Abstract.** Coupled physical-biogeochemical models can generally reproduce large-scale patterns of primary production and biogeochemistry, but they often underestimate observed variability and gradients. This is partially caused by insufficient representation of systematic variations in the elemental composition and pigment density of phytoplankton. Although progress has been made through approaches accounting for the dynamics of phytoplankton composition with additional state variables, formidable computational challenges arise when these are applied in spatially explicit setups. The Instantaneous Acclimation (IA) approach addresses these challenges by assuming that Chl:C:nutrient ratios are instantly optimized locally (within each modelled grid cell, at each timestep), such that they can be resolved as diagnostic variables. Here we present the first tests of IA in an idealized, 1D setup: we implemented the IA in the Framework for Aquatic Biogeochemical Models (FABM), and coupled it with the General Ocean Turbulence Model (GOTM) to simulate the spatio-temporal dynamics in a 1-D water column. We compare the IA model against a fully dynamic, otherwise equivalently acclimative (DA) variant with an additional state variable, and a third, non-acclimative and fixed-stoichiometry (FS) variant. We find that, the IA and DA variants, which require the same parameter set, behave similarly in many respects, although some differences do emerge especially during the winter-spring and autumn-winter transitions. These differences however are relatively small in comparison to the differences between the DA and FS variants, suggesting that the IA approach can be used as a cost-effective improvement over a fixed stoichiometry approach. Our analysis provides insights into the roles of acclimative flexibilities in simulated primary production and nutrient drawdown rates, seasonal and vertical distribution of phytoplankton biomass, formation of thin chlorophyll layers and stoichiometry of detrital material.

## 1 Introduction

### 1.1 Modelling phytoplankton and their cellular composition:

In early ecosystem models, the elemental composition, i.e., proportion of carbon (C), nitrogen (N), and phosphorus (P) content of phytoplankton was generally assumed constant, and at least since the work of Dugdale (1967) their growth was typically described by the so-called 'Monod' model (Monod, 1949), which assumes a saturating response of the rate of carbon assimilation (and hence, of nutrient uptake) to the ambient nutrient concentration, described by a rectangular hyperbolic function. Similarly,

specific chlorophyll (Chl) content, i.e., Chl:C ratio was assumed to be constant, when comparing the simulated phytoplankton biomass against the *in-situ* or satellite based chlorophyll measurements. In many primary production modules coupled to general circulation models that are actively being used for various purposes to this date, phytoplankton C:Chl and/or C:N:P ratios are assumed to be constant (see e.g., the models in Laufkötter et al., 2015).

The inadequacy of these simplifying assumptions was made clear decades ago by the discovery that phytoplankton elemental composition (e.g., Gerloff and Skoog, 1954) and chlorophyll content (e.g., Platt and Jassby, 1976) are variable. Chl:C:N:P ratios of phytoplankton have since been found to vary widely in many laboratory experiments (e.g., Kruskopf and Flynn, 2006) and field observations (e.g., Martiny et al., 2013; Burson et al., 2016). Since the work of Caperon (1968) and Droop (1968), the so called 'quota' (or variable internal stores, or 'Droop') model has been widely employed to describe the dynamics of carbon and nutrients bound to phytoplankton, using a separate state variable for each element or nutrient resolved. For describing variable Chl:C ratios, acclimation models, most commonly that of Geider et al. (1998), but also others (e.g. Pahlow and Oschlies, 2009; Wirtz and Kerimoglu, 2016) are being increasingly employed in biogeochemical model frameworks. Such models typically couple a description of variable N:C (or other nutrient:C) with photoacclimation, i.e., variation of Chl:C, using one more state variable for Chl bound to phytoplankton (Moore et al., 2002; Schourup-Kristensen et al., 2014; Kerimoglu et al., 2017; Kwiatkowski et al., 2018). Some models assume a constant N:C ratio, while describing the variations in Chl:C, e.g., using only the photoacclimation portion (e.g., Moore et al., 2004) of the model by Geider et al. (1998) or using an empirical function (e.g., Oschlies and Schartau, 2005), that was earlier proposed by Cloern et al. (1995).

Models that account for variations in cellular composition are in principle more likely to provide more realistic estimates of phytoplankton biomass and biogeochemical fluxes: when the variabilities in Chl:C and C:nutrient ratios are realistically represented by the models, their calibration on the basis of in situ and satellite Chl observations become more accurate (Behrenfeld et al., 2009; Ayata et al., 2013; Kerimoglu et al., 2017), and their estimates of biosynthesis rates of C and nutrients, consequently the drawdown of nutrients, and elemental composition of the export flux can be better reproduced (Anderson and Pondaven, 2003; Mongin et al., 2003) respectively. However, the mechanistic basis of some of the models remain questionable, given their parameterization of certain processes using heuristic or empirically inspired functions (Flynn et al., 2015). Moreover, schemes that require additional state variables, due to the need to calculate their transport as tracers, impose substantial computational costs. Especially for models that contain many phytoplankton functional types or clones (e.g., 350 in Dutkiewicz et al., 2020), such additional computational costs may severely limit the kinds of simulations and *in silica* experiments that can be conducted.

## 1.2 An optimality based resource allocation model

For the prediction of growth, nutrient uptake and acclimative variations of pigment and nutrient content of phytoplankton in response to changes in resource environment, such as the availability of mineral nutrients and light, 'Resource Allocation Models' (RAM) have been used (Shuter, 1979; Laws and Chalup, 1990; Armstrong, 1999; Klausmeier et al., 2004; Pahlow, 2005; Wirtz and Kerimoglu, 2016). This approach is based on the expectation that evolution has produced organisms that strive to maximize their net growth rate by optimally allocating their resources to cellular functions. The dependence of all

such functions on common resources therefore implies eco-physiological trade-offs (Smith et al., 2011). In this study, we specifically consider four physiological variables for describing the acclimative flexibilities involved in phytoplankton growth, as described by Pahlow et al. (2013):

$Q$: N quota (i.e., N:C ratio [molN molC$^{-1}$]) of phytoplankton.

$f_{\mathbf{V}}$: fractional allocation (dimensionless) to the nutrient uptake compartment (protoplast) to optimize the trade-off between photosynthesis ($\mu$) and nutrient uptake ($V$), as described by Pahlow and Oschlies (2013).

$f_{\mathbf{A}}$: fractional allocation (dimensionless) to affinity to optimize the trade-off between nutrient affinity ($A$) and maximum uptake rate within the nutrient uptake compartment ($V_{\max}$), as described by Pahlow (2005) and Smith et al. (2009).

$\hat{\theta}$: Chl:C ratio in chloroplasts ($\hat{\theta}$, [gChl molC$^{-1}$]) to optimize the trade-off between energy gained by light harvesting and energetic costs of chlorophyll synthesis and maintenance, as described by Pahlow et al. (2013).

## 1.3 Instantaneous Acclimation (IA) approach

As in most previous models of flexible phytoplankton composition, the above mentioned model by Pahlow et al. (2013) explicitly resolved the dynamics of the carbon, nitrogen, and chlorophyll within phytoplankton biomass. This approach is well suited for simulating the short-term (i.e., hours to days) dynamics of growth, and hence for testing model assumptions against the results of batch culture experiments (e.g., Pahlow, 2005; Pahlow et al., 2013). Resolving the transient dynamics is important for such short-term experiments, where the response of phytoplankton may differ substantially in terms of nutrient uptake versus carbon-based growth and chlorophyll synthesis.

By contrast, oceanic (or even freshwater) observations are rarely available at such fine temporal resolution. The lack of observations at sufficient temporal resolution to test short-term model dynamics motivated the development of the Instantaneous Acclimation (IA) approach (Smith et al., 2016) as a way to potentially capture growth response at longer timescales while requiring substantially fewer calculations. IA is based on the balanced growth assumption, which Burmaster (1979) showed was able to reconcile the ability of the Droop, Monod and Michaelis-Menten models to capture phytoplankton growth response at steady state, as measured by continuous culture experiments. The key assumption is that growth and nutrient uptake are at all times strictly balanced with respect to the internal C:N stoichiometry of the cell (see Sect. 2.2 below for details). Based on this assumption, IA calculates only one specific rate for both growth and nutrient uptake. Smith et al. (2016) applied this assumption in a 0-D (box) model, adequate for reproducing sparse oceanic observations, but did not evaluate its performance compared to fully dynamic models of flexible composition.

Ward (2017) compared the results of a phytoplankton model with instantaneosly adjusting quota against a fully dynamic model with explicit state variables for each element resolved, and a fixed stoichiometry model, in a 0-D setup. He found that for a wide range of realistic forcing dynamics, the instantaneous approach yielded results practically indistinguishable from the fully dynamic model whereas these results differed considerably from those of the fixed stoichiometry model. To our knowledge, the IA approach has yet to be tested in a spatially explicit model, where the inclusion of transport terms may lead

to additional complications. In a spatially structured environment, transport of cells with a certain internal state to a zone where the typical (average) cellular composition differs, can result in a spatial storage advantage (Grover, 2009). A typical example of this is nutrient-replete cells (as represented by high N:C) at the deeper layers diffusing towards the Surface Mixed Layer (SML) across the thermocline where the cells are typically nutrient starved (e.g., Kerimoglu et al., 2012). In principal, this effect can be resolved only by explicitly tracing the constituents of the cell dynamically.

## 1.4  Objectives of this study

This study presents a novel implementation of the IA-approach in the Framework for Aquatic Biogeochemical Models (FABM Bruggeman and Bolding, 2014) , and an assessment of its behavior compared to two other established variants (Fig. 1): the first is the widely used, non-acclimative, **Fixed-Stoichiometry (FS)** variant, which resolves only the N bound to phytoplankton explicitly. The second variant is the **Dynamic Acclimation (DA)** variant, which resolves the C and N bound to phytoplankton

fully dynamically, with two state variables. The comparisons of the three model variants were conducted to answer the following two specific questions: *(i)* how do the simulations performed with the IA variant differ from those of the fully dynamic DA variant? and *(ii)* compared to the FS variant, do the results of the IA variant differ substantially? While answering these questions, we aimed to gain mechanistic understanding of the dynamics driving the difference between the model results.

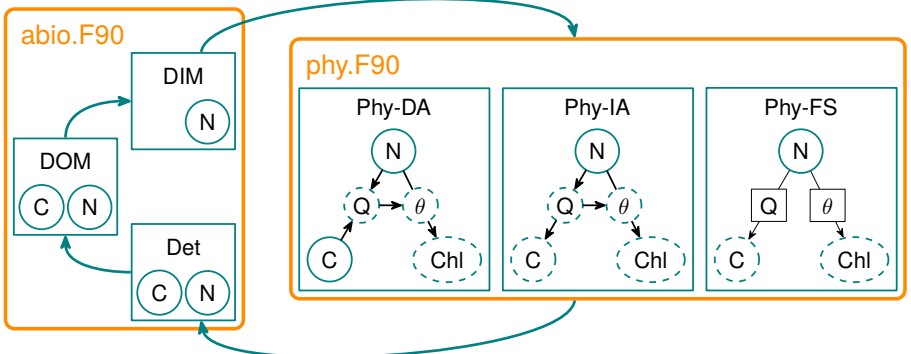

**Figure 1.** Diagram of the FABM-NflexPD model. Abitoic components, $DOM$, $DIM$ and $Det$ are calculated by the module `abio.F90`, which are then coupled to the phytoplankton simulated by the module `phy.F90` that simulates the dynamics of $Phy_N$, $Phy_C$ and $Phy_{Chl}$ by the DA, IA and FS variants (see section 2.2.2). Solid circles in the phytoplankton module represent state variables, dashed circles/ellipsoids represent diagnostically calculated variables and solid squares (for FS) represent prescribed values. The DA variant estimates the N, C and Chl content of phytoplankton based on a resource allocation scheme, whereas the FS variant estimates only N prognostically, while C and Chl are based on prescribed values of nitrogen quota ($Q$) and cellular Chl:C ratio ($\theta$) (see the text).

In the following sections, we describe the general structure of the model, the details of the physiological flexibilities men-

tioned above for each model variant, and the setup to simulate the model. Then we show the results of the simulated patterns of phytoplankton in terms of carbon, nitrogen, and chlorophyll, cell quota ($Q$), Chl:C ratio, as well as the fractional allocations. We finally discuss the advantages, as well as the challenges and limitations of implementing the IA approach.

## 2 Model Description

### 2.1 General Structure

For describing the cycling of N, we consider a simple model structure (Fig. 1) with four compartments: C and N bound to phytoplankton ($Phy_C$, $Phy_N$), detritus ($Det_C$, $Det_N$), dissolved organic matter ($DOC$, $DON$) and Dissolved Inorganic Nitrogen ($DIN$). Note that our model does not resolve the dynamics of Dissolved Inorganic Carbon, $DIC$ per se.

The coupled set of differential equations ($s(x)$ short for $\frac{dx}{dt}$) that describe the dynamics of state variables are provided in Eqs. (1–4), where each of the coupled C/N terms are annotated with the processes they represent. The formal definition and exact formulation of the flux terms ($F_{\text{FROM-TO}}$) in Eqs. (1–4) that are trivial (i.e., all except $F_{DIN-Phy_N}$ and $F_{DIC-Phy_C}$) are
115 provided in Table 1. For equations applying only to a subset of our model variants, the variants are indicated near the equation number in curly braces ({}). In addition, Table 2 provides an overview of how the model variants differ.

$$s(Phy_N) = F_{DIN-Phy_N} - F_{Phy_N-Det_N} \tag{1a}$$

$$s(Phy_C) = \underbrace{F_{DIC-Phy_C}}_{\text{Uptake}} - \underbrace{F_{Phy_C-Det_C}}_{\text{Mortality}} \qquad \{\text{DA}\} \tag{1b}$$

$$s(Det_N) = F_{Phy_N-Det_N} - F_{Det_N-DON} \tag{2a}$$

$$s(Det_C) = \underbrace{F_{Phy_C-Det_C}}_{\text{Mortality}} - \underbrace{F_{Det_C-DOC}}_{\text{Hydrolysis}} \tag{2b}$$

$$s(DON) = F_{Det_N-DON} - F_{DON-DIN} \tag{3a}$$

$$s(DOC) = \underbrace{F_{Det_C-DOC}}_{\text{Hydrolysis}} - \underbrace{F_{DOC-DIC}}_{\text{Remineralization}} \tag{3b}$$

$$s(DIN) = \underbrace{F_{DON-DIN}}_{\text{Remineralization}} - \underbrace{F_{DIN-Phy_N}}_{\text{N uptake}} \tag{4}$$

It should be noted that the $Phy_C$ is resolved as a state variable only by the DA variant (Eq. (1b)). The terms $F_{DIN-Phy_N}$ and $F_{DIC-Phy_C}$ have central importance to this study and deserve explanation. $F_{DIN-Phy_N}$ represents the net N flux from the
130 DIN to phytoplankton, and is given by the product of the phytoplankton carbon biomass, $Phy_C$ and the specific nutrient uptake rate, $V$:

$$F_{DIN-Phy_N} = V \cdot Phy_C \tag{5}$$

For the FS and IA variants, balanced growth (Burmaster, 1979) is assumed, such that $V$ is directly linked to net growth rate, $\mu$, via the nutrient quota, $Q$:

$$V = \mu \cdot Q \qquad \{\text{FS, IA}\} \tag{6}$$

whereas for the DA variant, $V$ is calculated explicitly (Eq. (12)). Net growth rate, $\mu$, is obtained by subtracting the respiration costs associated with chlorophyll maintenance and synthesis, $R_{\text{Chl}}$, and nutrient uptake, $R_{\text{N}}$, from the cellular gross growth rate, $\mu_{\text{g}}$ (Eq. (13)):

$$\mu = \mu_{\text{g}} - R_{\text{Chl}} - R_{\text{N}} = \mu_{\text{net}} - \zeta_{\text{N}} \cdot V \tag{7}$$

where $\zeta_{\text{N}}$ is the cost of N assimilation (Table 3) and $R_{\text{Chl}}$ is the cost of chlorophyll synthesis and maintenance (Section 2.2.4).

$F_{DIC-Phy_{\text{C}}}$ is required only by the DA variant that explicitly resolves the dynamics of $Phy_{\text{C}}$ (Eq. (1b)). It is given by the product of net growth rate, $\mu$ with $Phy_{\text{C}}$, as is typical in quota models (Caperon, 1968; Droop, 1968):

$$F_{DIC-Phy_{\text{C}}} = \mu \cdot Phy_{\text{C}} \tag{\{DA\} (8)}$$

**Table 1.** Definitions, expansions/values and units of terms/symbols regarding the fluxes between model compartments.

| Term/Symbol | Definition | Expansion/Value | Units |
|---|---|---|---|
| $F_{Phy_{\text{N}}-Det_{\text{N}}}$ | N flux from $Phy_{\text{N}}$ to $Det_{\text{N}}$ | $m \cdot Phy_{\text{N}}^2$ | $\text{mmolN m}^{-3}\text{d}^{-1}$ |
| $F_{Phy_{\text{C}}-Det_{\text{C}}}$ | C flux from $Phy_{\text{C}}$ to $Det_{\text{C}}$ | $F_{Phy_{\text{N}}-Det_{\text{N}}}/Q$ | $\text{mmolC m}^{-3}\text{d}^{-1}$ |
| $F_{Det_{\text{N}}-DON}$ | N flux from $Det_{\text{N}}$ to $DON$ | $r_{\text{hyd}} \cdot Det_{\text{N}}$ | $\text{mmolN m}^{-3}\text{d}^{-1}$ |
| $F_{Det_{\text{C}}-DOC}$ | C flux from $Det_{\text{C}}$ to $DOC$ | $r_{\text{hyd}} \cdot Det_{\text{C}}$ | $\text{mmolC m}^{-3}\text{d}^{-1}$ |
| $F_{DON-DIN}$ | N flux from $DON$ to $DIN$ | $r_{\text{rem}} \cdot DON$ | $\text{mmolN m}^{-3}\text{d}^{-1}$ |
| $F_{DOC-DIC}$ | C flux from $DOC$ to $DIC$ | $r_{\text{rem}} \cdot DOC$ | $\text{mmolN m}^{-3}\text{d}^{-1}$ |
| $m$ | Mortality rate coefficient | 0.1 | $\text{m}^3\text{mmolN}^{-1}\text{d}^{-1}$ |
| $r_{\text{hyd}}$ | Hydrolysis rate constant | 0.1 | $\text{d}^{-1}$ |
| $r_{\text{rem}}$ | Remineralization rate constant | 0.1 | $\text{d}^{-1}$ |

## 2.2    Flexibilities Represented by the Model Variants

We compare the behavior of three model variants that differ in their representation of the physiological flexibilities. These variants are:

**'Dynamic Acclimation' (DA):** explicitly describes the dynamics of nitrogen and carbon bound to phytoplankton, and the acclimation mechanisms introduced in Section 1.2, here as represented by flexibilities in growth vs. nutrient uptake; nutrient affinity vs. maximum uptake; and chlorophyll density in chloroplasts; each of which are explained in detail in 150      the following sections. A full description of this variant (including diazotrophy) can be found in Pahlow et al. (2013).

**'Instantaneous Acclimation' (IA):** assumes that the nitrogen quota (molar N:C ratio) adjusts instantaneously to its optimal value locally (i.e., at any point in time and space), but is otherwise identical to the DA variant with respect to the acclimation mechanisms. A full description of this variant can be found in Smith et al. (2016).

**'Fixed Stoichiometry' (FS)** which assumes no physiological acclimation or quota variability whatsoever.

In the following, representations of the acclimative flexibilities by each model variant are explained in detail.

### 2.2.1 Flexibility I: Nutrient Quota

Flexibility in the elemental composition of phytoplankton (Q) is a result of acclimation processes, such as synthesis of enzymes or pigments, which differ in elemental composition (e.g. Geider and La Roche, 2002), in response to changes in resource (light and nutrients) availability.

**DA:** For the dynamic-acclimation variant, $Q$, is the ratio of the phytoplankton N and C state variables:

$$Q = \frac{Phy_N}{Phy_C} \qquad\qquad \text{\{DA\} (9)}$$

**IA:** For the instantaneous-acclimation variant, $Q$ is assumed to adjust instantaneously to its balanced-growth optimum ($Q^o$) according to Pahlow and Oschlies (2013):

$$Q^o = \frac{Q_0}{2}\left[1 + \sqrt{1 + \frac{2}{Q_0(\hat{\mu}_{net}/\hat{V} + \zeta_N)}}\right] \qquad\qquad \text{\{IA\} (10)}$$

where, $\hat{\mu}_{net}$ and $\hat{V}$ are the chloroplast-specific net growth and protoplast-specific N uptake rates (Table 2), and $Q_0$ and $\zeta_N$ and are the subsistence quota, and cost of N uptake (Table 3), respectively. Note that this solution differs slightly from the solution proposed by Smith et al. (2016), where the cost of chlorophyll maintenance and synthesis was ignored. (see Appendix 1 for details).

**FS:** In the fixed-stoichiometry variant, $Q$ is a prescribed parameter (Table 2).

### 2.2.2 Flexibility II: growth vs nutrient uptake

Given the high nitrogen content in the enzymes responsible for both $CO_2$ fixation and nutrient uptake and assimilation (Geider and La Roche, 2002), we consider a trade-off in the allocation of nitrogen between carbon fixation and nutrient uptake for the acclimative variants, whereas this trade-off is ignored for the FS variant.

**DA & IA:** For the acclimative variants, following Pahlow and Oschlies (2013), the trade-off is specified in terms of the fraction of cellular nitrogen reserves allocated to nitrogen uptake ($f_V$), which linearly increases $V$, and decreases $\mu_g$, through decreasing the resources available for carbon fixation, $f_C$, which is interpreted as the relative size of the chloroplast (Pahlow and Oschlies, 2013).

$$f_C = \left(1 - \frac{Q_0}{2Q} - f_V\right) \qquad\qquad \text{\{IA, DA\} (11)}$$

where $f_V$ is the fractional allocation towards nutrient uptake for the DA variant (see Eq. (6) for IA variant):

$$V = f_V \cdot \hat{V} \qquad\qquad \text{\{DA\} (12)}$$

**Table 2.** Summary of differences between model variants. NA denotes not applicable. (*): prescribed, see Table 3.

| Term | Definition | Equation/Definition IA | DA | FS | Units |
|------|-----------|------|------|------|------|
| $Phy_C$ | Carbon bound to phytoplankton | $Phy_N/Q$ | Eq. (1)b | $Phy_N/Q$ | mmolC m$^{-3}$ |
| $F_{DIN-Phy_N}$ | N flux from DIN to $Phy_N$ | | Eq. (5) | | mmolN m$^{-3}$ d$^{-1}$ |
| $F_{DIC-Phy_C}$ | C flux from DIC to $Phy_N$ | NA | Eq. (8) | NA | mmolc m$^{-3}$ d$^{-1}$ |
| $f_V$ | Fractional allocation to uptake | | Eq. (14) | (*) | – |
| $V$ | Specific N uptake rate | Eq. (6) | Eq. (12) | Eq. (6) | molN molC$^{-1}$ d$^{-1}$ |
| $\mu$ | Cellular net growth rate | | Eq. (7) | | d$^{-1}$ |
| $\mu_g$ | Cellular gross growth rate | | Eq. (13) | | d$^{-1}$ |
| $\mu_{net}$ | $\mu_g - R_{Chl}$ | | Eq. (7) | | d$^{-1}$ |
| $\hat{\mu}_g$ | Gross growth rate within chloroplast | | Eq. (21) | | d$^{-1}$ |
| $\hat{\mu}_{net}$ | Net growth rate within chloroplast | | $\hat{\mu}_g - \hat{R}_{Chl}$ | | d$^{-1}$ |
| $Q$ | N quota | Eq. (10) | $Phy_N/Phy_C$ | (*) | molN molC$^{-1}$ |
| $\hat{V}$ | Protoplast-specific N uptake rate | | Eq. (16) | | molN molC$^{-1}$ d$^{-1}$ |
| $f_A$ | Fractional allocation to affinity | | Eq. (18) | (*) | – |
| $\hat{\theta}$ | Chl:C in chloroplasts | | Eq. (26) | (*) | gChl molC$^{-1}$ |
| $\theta$ | Chl:C in the entire cell | | Eq. (24) | | gChl molC$^{-1}$ |
| $R_N$ | Cost of N uptake | | Eq. (7) | | d$^{-1}$ |
| $\hat{R}_{Chl}$ | Cost of light harvesting within chloroplast | | Eq. (23) | | d$^{-1}$ |
| $R_{Chl}$ | Cellular cost of light harvesting | | Eq. (25) | | d$^{-1}$ |
| $f_C$ | Fractional N allocation to C-fixation | | Eq. (11) | NA | – |
| $L_N$ | Nutrient limitation term | | NA | Eq. (15) | – |
| $L_I$ | Light limitation | | Eq. (22) | | – |

where $\hat{V}$ is the protoplast-specific N uptake rate (see below). The cellular gross growth rate is then determined by scaling the gross growth rate within the chloroplast $\hat{\mu}_g$ (see Section 2.2.4) by the relative size of the chloroplast, $f_C$:

$$\mu_g = f_C \cdot \hat{\mu}_g \qquad (13)$$

Note that, for calculating the effective flux from DIN to $Phy_N$ (Eq. (5)), only the DA variant uses $V$ as calculated by Eq. (12), while the IA variant calculates the uptake rate from the growth rate, based on the balanced growth assumption (Eq. (6)). However, the IA variant still needs the $V$ as calculated by Eq. (12), for calculating the costs of nutrient uptake (Eq. (7)).

Both acclimative variants assume that $f_V$ maximises the net specific growth rate under balanced growth conditions. Following Pahlow and Oschlies (2013), this optimal value is found as (see Appendix 1):

$$\frac{d\mu}{df_V} = 0 \quad \Rightarrow \quad f_V = \left(\frac{Q_0}{2Q}\right) - \zeta_N(Q - Q_0) \qquad \text{\{IA, DA\} (14)}$$

**FS:** For the fixed-stoichiometry variant, the gross growth rate, $\mu_g$ is obtained by the multiplication of $\hat{\mu}_g$, for FS, interpreted as the light-limited potential growth rate, with a nutrient limitation term $L_N$, formulated as a hyperbolic function of ambient DIN concentration, following the Michaelis-Menten-Monod model (Johnson and Goody, 2011; Monod, 1949):

$$\mu_g = \hat{\mu}_g \cdot L_N = \hat{\mu}_g \cdot \frac{DIN}{K_N + DIN} \qquad \text{\{FS\} (15)}$$

Thus, for the FS variant, $\mu$ (Eq. (7)), and hence, through the balanced growth assumption, $V$ (Eq. (6)) are directly linked to the external nutrient concentration (Eq. (15)) as in typical fixed-stoichiometry models. Given the fact that both $L_N$ (Eq. (15)) for the FS variant and $f_C$ (Eq. (11)) for the acclimative variants have an equivalent role (in scaling $\hat{\mu}_g$ to $\mu_g$), and they both represent nutrient limitation, we consider them to be comparable, i.e., $L_N \sim f_C$.

### 2.2.3 Flexibility III: nutrient affinity vs. maximum uptake rate

**DA & IA:** Originally introduced for describing the substrate uptake by bacteria, 'affinity' of a microorganism 'can be viewed as a measure of effective collusion between substrate and transport site' (Button, 1978), which can be practically found from the initial slope (i.e., before saturation) of the uptake rate with respect to the substrate concentration (Button, 1978). The term has been used for describing the nutrient uptake by phytoplankton (Aksnes and Egge, 1991), and recognized to be a measure of competitive ability under low concentrations. The maximum nutrient uptake rate, on the contrary, can taken to be a measure of competitiveness under hight nutrient concentrations. The protoplast-specific N uptake rate, $\hat{V}$ can be described by a function of maximal uptake rate, $\hat{V}_{\max}$, and nutrient affinity, $\hat{A}$:

$$\hat{V} = \frac{\hat{V}_{\max} \cdot \hat{A} \cdot DIN}{\hat{V}_{\max} + \hat{A} \cdot DIN} \qquad \text{\{IA, DA\} (16)}$$

The acclimation variants introduce a trade-off between affinity vs. maximum uptake rate. This trade-off is captured by the fractional allocation of resources to affinity ($f_A$), which increases affinity, $\hat{A} = f_A \hat{A}_0$, while decreasing maximum uptake rate, $\hat{V}_{\max} = (1 - f_A)\hat{V}_0$, so that Eq. (16) becomes:

$$\hat{V} = \frac{(1 - f_A)\hat{V}_0 \cdot f_A \hat{A}_0 \cdot DIN}{(1 - f_A)\hat{V}_0 + f_A \hat{A}_0 \cdot DIN} \qquad \text{\{IA, DA\} (17)}$$

$f_A$ is set to its optimum value, which maximizes $\hat{V}$, and hence also $V$ (Pahlow, 2005):

$$\frac{d\hat{V}}{df_A} = 0 \quad \Rightarrow \quad f_A = \left[1 + \sqrt{\frac{\hat{A}_0 \cdot DIN}{\hat{V}_0}}\right]^{-1} \qquad \text{\{IA, DA\} (18)}$$

**FS:** The fixed-stoichiometry variant ignores this trade-off entirely, by describing the nutrient limitation with the Michaelis-
Menten-Monod function (Eq. (15)). Following Button (1978) and Smith et al. (2009), the $K_N$ parameter in Eq. (15), can
be expressed as a function of $V_{\max}$ and $\hat{A}$, according to:

$$K_N = \frac{\hat{V}_{\max}}{\hat{A}} = \frac{(1 - f_A) \cdot \hat{V}_0}{f_A \cdot \hat{A}_0} \qquad \{FS\} \quad (19)$$

Based on Eq. (19), corresponding $K_N$ values were diagnosed from the solution of the IA variant (i.e., using the locally
optimized $f_A$ values as calculated with Eq. (18), and $\hat{A}_0$ and $\hat{V}_0$ parameters specified for the IA and DA variants). The
biomass-weighted spatio-temporal average $K_N$ value so obtained was prescribed for the FS variant (Table 3).

### 2.2.4 Flexibility IV: photoacclimation

Photo-acclimation is based on the net carbon fixation rate within the chloroplast, $\hat{\mu}_{net}$ (equivalent to $\mathscr{A}$ in Pahlow and Oschlies
(2013)), which is obtained by subtracting the chloroplast specific synthesis and maintenance costs of chlorophyll, from the
gross growth rate within the chloroplast, i.e.,

$$\hat{\mu}_{net} = \hat{\mu}_g - \hat{R}_{Chl} \qquad (20)$$

where, $\hat{\mu}_g$ is given by the product of daylength as a fraction of 24 h,, $L_D$, potential turnover rate, $\hat{\mu}_0$, and the light-saturation of
the photosynthetic apparatus, $L_I$:

$$\hat{\mu}_g = L_D \hat{\mu}_0 L_I \qquad (21)$$

$L_I$ is a saturating function of daytime average light, $\bar{I}$, and chlorophyll density in chloroplasts, $\hat{\theta}$:

$$L_I = 1 - \exp\left(\frac{-\alpha \hat{\theta} \bar{I}}{\hat{\mu}_0}\right) \qquad (22)$$

where $\alpha$ is light affinity. Returning to Eq. (20), $\hat{R}_{Chl}$ is given by:

$$\hat{R}_{Chl} = \left(\hat{\mu}_g + R_M^{Chl}\right) \zeta_{Chl} \hat{\theta} \qquad (23)$$

where $R_M^{Chl}$ and $\zeta_{Chl}$ are the costs of chlorophyll maintenance and synthesis, respectively (Table 3).

Photo-acclimation is mainly represented in terms of the chlorophyll density in chloroplasts, $\hat{\theta}$. Increasing $\hat{\theta}$ reduces light
limitation (Eq. (22)), but at the expense of greater respiration costs (Eq. (23)). In turn, for obtaining the cellular Chl:C ratio, $\theta$,
is calculated by multiplying $\hat{\theta}$ times $f_C$, i.e., size of the chloroplast:

$$\theta = f_C \cdot \hat{\theta} \qquad \{IA,DA\} \quad (24)$$

Similarly, the overall respiratory cost of maintaining cellular chlorophyll is obtained by multiplying the chloroplast-specific
cost by the size of the chloroplast:

$$R_{Chl} = f_C \cdot \hat{R}_{Chl} \qquad \{IA,DA\} \quad (25)$$

Although $\theta$ (Eq. (24)) is only a diagnostic quantity, $R_{\text{Chl}}$ (Eq. (25)) directly determines the net growth rate through Eq. (7). Therefore, scaling of the chloroplast-specific respiration rate, $\hat{R}_{\text{Chl}}$ by $f_{\text{C}}$ can considered to be an acclimative quality implied by variable $f_{\text{V}}$ and $Q$, which, in combination (Eq. (11)), determine the chlorophyll maintenance cost through Eq. (25)).

**IA & DA:** In the acclimation variants, $\hat{\theta}$ is assumed to adjust instantaneously to its optimal value, which maximizes $\hat{\mu}_{\text{net}}$. Following Pahlow et al. (2013) this optimal value is:

$$\hat{\theta} = \begin{cases} \dfrac{1}{\zeta_{\text{Chl}}} + \dfrac{\hat{\mu}_0}{\alpha \bar{I}}\left(1 - W_0\left[\left(1 + \dfrac{R_{\text{M}}^{\text{Chl}}}{L_{\text{D}}\hat{\mu}_0}\right)\exp\left(1 + \dfrac{\alpha \bar{I}}{\hat{\mu}_0 \zeta_{\text{Chl}}}\right)\right]\right), & \bar{I} > \bar{I}_{\text{C}} \\ 0, & \bar{I} \leq \bar{I}_{\text{C}} \end{cases} \qquad \{\text{IA, DA}\} \quad (26)$$

where $W_0$ is the 0-branch of the Lambert-W function, $\bar{I}$ is the daytime average irradiance (i.e., $\hat{I} = \bar{I}_{24\text{h}}/L_{\text{D}}$) and $\bar{I}_{\text{C}}$ is the critical daytime average irradiance level, above which chlorophyll synthesis is worthwhile (Pahlow et al., 2013):

$$\bar{I}_{\text{C}} = \frac{\zeta_{\text{Chl}} R_{\text{M}}^{\text{Chl}}}{\alpha L_{\text{D}}} \qquad (27)$$

**FS:** For the fixed-stoichiometry variant, $\hat{\theta}$ is prescribed as the biomass-weighted average value calculated by the IA variant. Considering that $\theta$ is typically a constant 'conversion factor' in classical, fixed stoichiometry and fixed Chl:C models, in Eqs. (24) and (25), we assume that the size of the chloroplast, $f_{\text{C}}$, is constant too. For the sake of consistency with the IA variant, $f_{\text{C}}$ for FS is diagnosed from its expanded form, i.e., $1 - \frac{Q_0}{2Q} - f_{\text{V}}$ (Eq. (11)). Hence, in addition to the prescribed value of $Q$ (see Section 2.2.1), the biomass-weighted mean of $f_{\text{V}}$, as calculated by the IA variant is prescribed (Table 3). Given the comparability of the terms (Section 2.2.2), diagnosing $f_{\text{C}}$ from $L_{\text{N}}$ comes into question, which is elaborated in Appendix B.

### 2.2.5 Temperature Scaling

Kinetic rate constants: $m$, $r_{\text{hyd}}$, $r_{\text{rem}}$ in Table 1, and $\hat{V}_0$, $\hat{A}$, $\hat{A}_0$ and $R_{\text{M}}^{\text{Chl}}$ in Table 3 are prescribed for a reference temperature of $T_{\text{r}} = 20\,°\text{C} = 293.15\,\text{K}$, and scaled to the ambient temperature in water, $T$ (in K), according to the Arrhenius function:

$$f(T) = \exp\left(\frac{-E_{\text{a}}}{R}\left[\frac{1}{T} - \frac{1}{T_{\text{r}}}\right]\right) \qquad (28)$$

where the gas constant $R = 8.3145\,\text{J}\,\text{mol}^{-1}\text{K}^{-1}$, and the activation energy, $E_{\text{a}} = 4.82 \cdot 10^4\,\text{J}\,\text{mol}^{-1}$, such that every 10 degrees increase/decrease in $T$ approximately doubles/halves the reference rates.

### 2.3 Coupling with the Hydrodynamical Host

The model is implemented in the Framework for Aquatic Biogeochemical Models (i.e., FABM, Bruggeman and Bolding, 2014), so that it can be used, without modification, in combination with various hydrodynamical hosts. In this study, we performed simulations of an idealized water column, using the General Ocean Turbulence Model (i.e., GOTM Burchard et al.,

**Table 3.** Descriptions, values and units of model parameters regarding phytoplankton growth. Prescribed values for $Q$, $K_N$, $f_V$, and are based on the biomass-weighted averages estimated by the IA variant. All other parameter values are taken from within the published range (Pahlow et al., 2013; Smith et al., 2016), without particular reference to species.

| Term/Symbol | Definition | Value | Unit | Used by |
|---|---|---|---|---|
| $\hat{\mu}_0$ | Potential maximum growth rate | 5.0 | $\text{d}^{-1}$ | all |
| $Q_0$ | Subsistence quota | 0.039 | $\text{mmolN molC}^{-1}$ | IA, DA |
| $\hat{A}_0$ | Potential maximum nutrient affinity | 0.1 | $\text{m}^3 \text{ mmolC}^{-1} \text{ d}^{-1}$ | IA, DA |
| $\hat{V}_0$ | Potential maximum N uptake rate | 5.0 | $\text{molN molC}^{-1} \text{ d}^{-1}$ | IA, DA |
| $\alpha$ | Chl-specific slope of P-I curve | 1.0 | $\text{m}^2 \text{ E molCgChl}^{-1} \text{ d}^{-1}$ | all |
| $R_\text{M}^\text{Chl}$ | Cost of chlorophyll maintenance | 0.1 | $\text{d}^{-1}$ | all |
| $\zeta_\text{Chl}$ | Cost of chlorophyll synthesis | 0.5 | $\text{mmolC gChl}^{-1}$ | all |
| $\zeta_\text{N}$ | Cost of N uptake | 0.6 | $\text{molC molN}^{-1}$ | all |
| $Q$ | N quota | 0.084 | $\text{molN molC}^{-1}$ | FS |
| $K_\text{N}$ | Half saturation constant for N uptake | 4.84 | $\text{mmolN m}^{-3}$ | FS |
| $f_\text{V}$ | Fractional allocation to uptake | 0.32 | – | FS |
| $\hat{\theta}$ | Chl:C in chloroplasts | 0.518 | $\text{gChl molC}^{-1}$ | FS |

2006). GOTM calculates and provides the relevant physical quantities, such as $I$ (needed in Eq. (22)) and T (needed in Eq. (28)). $I$ is attenuated with depth (z) by various substances in water, according to:

$$I(z) = I_0 \left[ A \exp\left( \frac{-z}{\eta_1} \right) + (1 - A) \exp\left( \frac{-z}{\eta_2} - \int_z^0 \sum_i k_i c_i(z') \mathrm{d}z' \right) \right] \tag{29}$$

where $A$, $\eta_1$ and $\eta_2$ represent the differential attenuation length scales of red and blue light (Burchard et al., 2006), and $k_i$ is the specific attenuation coefficient of the biological quantities, which we set as $0.03 \text{ m}^2 \text{ mmolN}^{-1}$ for $Phy_\text{N}$ and $Det_\text{N}$. In order to account for background attenuation, we set the 'light extinction method' to 'Jerlov Type IB', corresponding to $A = 0.67$ $\eta_1 = 1.0 \text{ m}$, $\eta_2 = 17 \text{ m}$, characterizing water of medium clarity (Paulson and Simpson, 1977). Our results are qualitatively insensitive to these parameter settings. Besides providing necessary environmental variables, GOTM calculates the transport rates of the biological quantities, according to the general equation (Burchard et al., 2006):

$$\frac{\partial c_i}{\partial t} + \frac{\partial}{\partial z}\left( w_i c_i - K_z \frac{\partial c_i}{\partial z} \right) = s(c_i) \tag{30}$$

where, $K_\text{Z}$ is the eddy diffusivity calculated by GOTM, the source terms, $s(c_i)$ correspond to the Eqs. (1-4) and advection rates, $w_i$ are all set to 0.0, except that of detritus for which a sinking rate of -2.0 m d$^{-1}$ was specified. Note that the latter value was arbitrarily chosen to induce a downward flux in this idealized setup, and that in reality, it depends on the average size and density of detritus particles being modelled and displays a vast range (Guidi et al., 2008).

## 2.4 Idealized Setup and Simulations

We consider an idealized water column of 100 m depth. In order to mimic an environment that is characterized by strong seasonality, with deep mixed layers in spring and summer stratification, we force the model with astronomically calculated short wave radiation at 60°N latitude, and a repeating annual cycle of air temperature that ranges between 4–20 °C as described by a scaled sinusoidal function (Fig. 2).

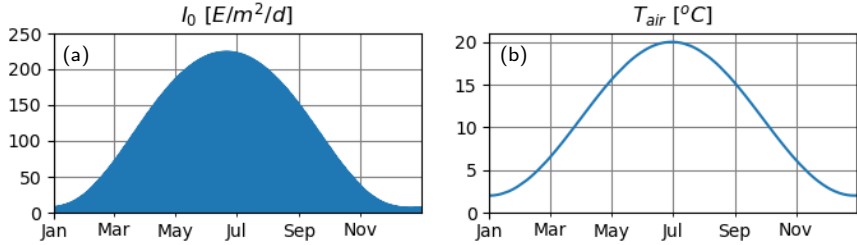

**Figure 2.** Atmospheric variables. (a) astronomically estimated instantaneus irradiance at the water surface, and (b) prescribed air temperature.

All other meteorological variables (wind speed, air pressure, humidity, and cloud cover) are assumed to be constant, and the model ignores precipitation and evaporation losses, as well as tidal variations. Starting from initial conditions, and annually repeating meteorological forcing as described above, each model variant was run for 3 years. The 3rd year results were nearly identical to those for the 2nd year, indicating that an equilibrium annual cycle was reached. In the following, we elaborate the seasonal dynamics during the 3rd year.

## 3 Results

Daytime-averaged irradiance, $\bar{I}$ and water temperature $T$ simulated by different model variants are very similar with subtle differences (Fig. 3a,d vs. b,e, vs. c,f), because each variant calculates slightly different phytoplankton biomass (see below), resulting in differences in attenuation of light and associated heating. Seasonal and vertical distributions of DIN as estimated by the model variants are similar (Fig. 3g-i). DIN depletion (<1 mmolN m$^{-3}$) during summer is confined to the upper 25 m as estimated by the FS variant, whereas it extends 5-10 m deeper as estimated by the IA and DA variants.

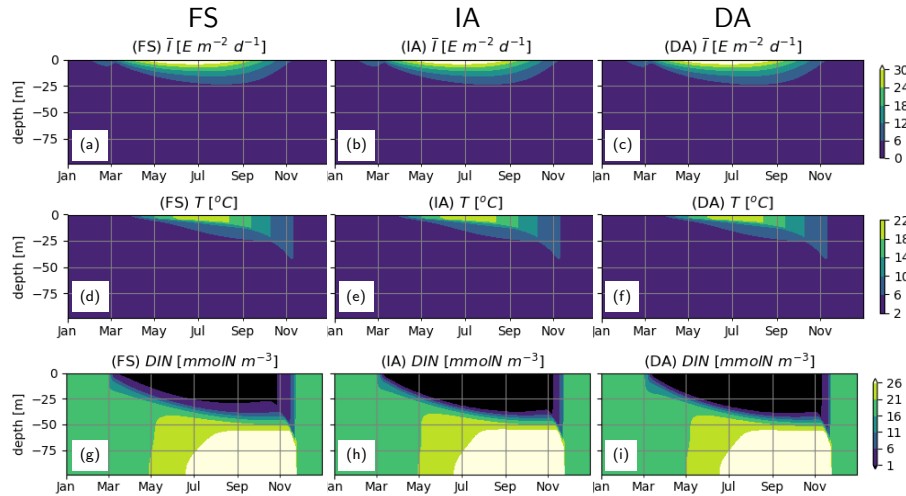

**Figure 3.** Abiotic environment. (a-c) daytime-averaged photosynthetically active radiation, $\bar{I}$ [E m$^{-2}$ d$^{-1}$], (d-f) water temperature $T$ [°C], and (g-i) DIN [mmolN m$^{-3}$], as simulated by the FS (left); IA (center) and DA (right) variants.

With all three model variants phytoplankton growth patterns are characterized by an intense surface bloom in spring, followed by a gradual deepening of the biomass maxima (Fig. 4a-c). Biomass concentration as estimated by the IA and DA variants during summer is greater than with the FS variant (Fig. 4a,b,c). Compared to the FS variant, the acclimation response in the other two variants tends to produce steeper gradients over both depth and time, because of combined dependencies on the three dynamically optimized allocation factors ($f_A$, $f_V$, and $\hat{\theta}$). This effect is most pronounced for $Phy_{Chl}$, which differs the most between the FS and the other two variants. With the FS variant, given the constant N:C ($Q$) and cellular Chl:C ($\theta$) (Fig. 4g,m), C, N and Chl bound to phytoplankton clearly display identical patterns (Fig. 4a,d,j; note that apparent differences in contour plots are due to contour limits not matching these ratios). IA and DA on the other hand simulate slightly different patterns for C, N and Chl bound to phytoplankton (Fig. 4b,e,k and c,f,l), because of the seasonally and vertically variable Chl:C:N. Decoupling of $Phy_N$ from $Phy_C$ is mainly monotonic, and is driven by increasing $Q$ with depth (Fig. 4h-i). On the other hand, decoupling of $Phy_{Chl}$ from $Phy_C$ follows a more complex pattern, because of the uni-modal distribution of $\theta$ across the water column (Fig. 4n-o). As a result of this uni-modality, Chl simulated by the IA and DA variants forms a distinct, thin layer below the thermocline (Fig. 4k-l).

During summer, $\hat{\theta}$ follows a complex, but roughly uni-modal distribution across depth (Fig. 5b-c): intermediate values at the surface first increase with depth to reach a maximum and then sharply decrease with increasing depth. The low values of $\hat{\theta}$ towards the surface reflect the optimization, which reduces pigment density when light is abundantly available because of the chloroplast-specific respiratory costs $\hat{\theta}$ (Eq. (23)). This can be seen in the flattening of the light-saturation function $L_I$ (Eq. (22)). In the deep layers, as $\bar{I}$ approaches $\bar{I}_C$, irradiance becomes insufficient to support the synthesis and maintenance of chlorophyll, and $\hat{\theta}$ rapidly converges to 0. $f_A$ and $f_V$ simulated by the IA and DA variants (Fig. 5e-f, h-i) increase with nutrient limitation (Fig. 5j-l) as expected (Smith et al., 2016). The fraction of resources available for carbon fixation, $f_C$, displays a

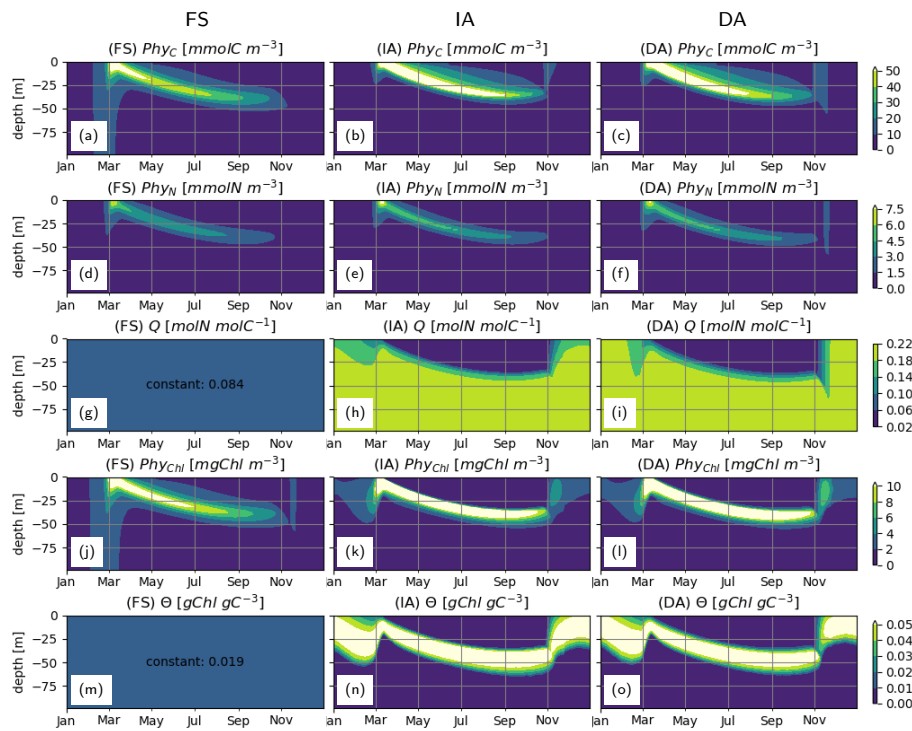

**Figure 4.** Phytoplankton C, N and Chl concentrations: (a-c) $Phy_C$ [mmolC m$^{-3}$], (d-f) $Phy_N$ [mmolN m$^{-3}$], (j-l) $Phy_{Chl}$ [mgC m$^{-3}$]; and phytoplankton N:C ($Q$) and Chl:C ($\Theta$) ratios: (g-i) $Q$ [molN molC$^{-1}$] and (m-o) $\Theta$ [gChl gC$^{-1}$], as simulated by the FS (left); IA (center) and DA (right) variants.

similar pattern in all model variants and is roughly the inverse of $f_V$: high during winter throughout the water column and in the deeper layers during summer, low in the upper layers during summer (Fig. 5j-l). For the FS variant, the pattern of the nutrient limitation term, $L_N$, is similar to the patterns of $f_C$ for IA and DA variants, Fig.5, although its magnitude in the summer is higher than other variants, as can be explained by the incomplete DIN depletion (Fig.3g, see below). Light saturation

of photosynthesis, $L_I$, displays a similar pattern in all variants (Fig. 5m-o), and mainly reflects irradiance levels (Fig. 3a-c). However, compared to the FS variant, the intermediate $L_I$ values in the IA and DA variants penetrate deeper (Fig. 5n,o vs m), because the optimization of $\hat{\theta}$ enhances light harvesting ability at these intermediate depths (Fig. 5b,c).

During winter and spring blooms, the net cellular growth rate, $\mu$ as estimated by the FS variant temporarily exceeds those estimated by the acclimative variants (Fig. 6a-c, see below for the explanation). The IA and DA variants estimate higher nutrient

uptake rates, $V$, in surface layers during the spring bloom, and in deeper layers during summer (Fig. 6d-f). Negative $V$ in the bottom layers as estimated by the FS and IA variants is a direct result of the balanced growth assumption (Eq. (6)) and can be interpreted as exudation. Respiratory costs of nutrient uptake, $R_N$, (Fig. 6h-i) are much lower than $R_{Chl}$ (Fig. 6j-l). For the FS variant, $R_N$ drops below 0 in the deeper (>50m) waters, implying negative respiration, which is a model artefact, as a result of $\hat{\mu}_{net}$ becoming negative (see Eq. (A4) in Sect.A1) due to the fixed $\hat{\theta}$. However these negative values are small, and therefore

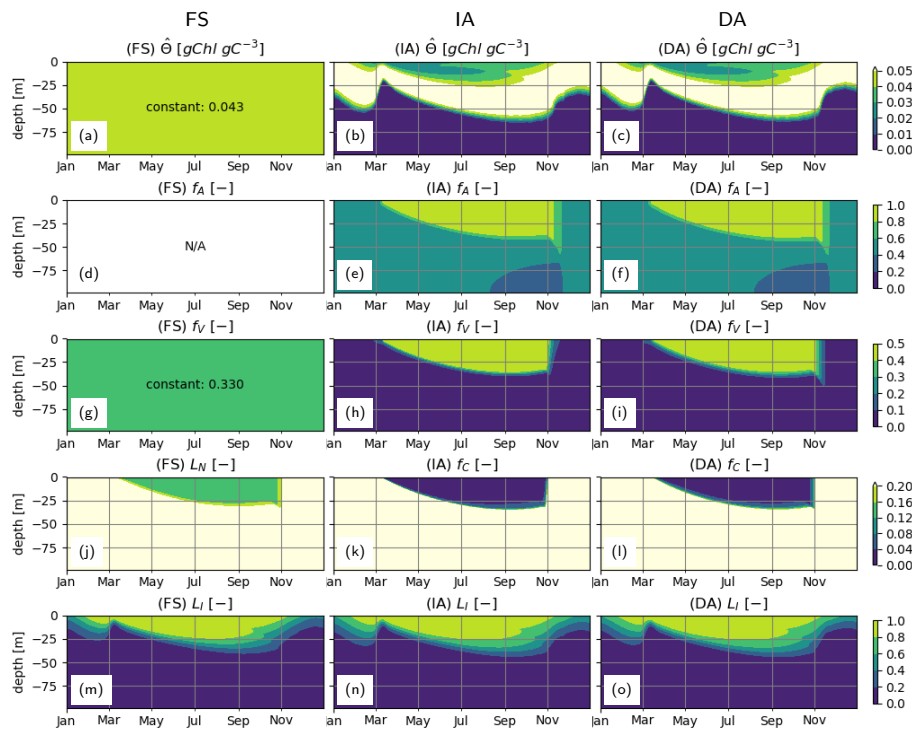

**Figure 5.** Phytoplankton physiological variables. (a-c) Chlorophyll density in chloroplasts, $\hat{\Theta}$ [gChl gC$^{-1}$]; (d-f) fractional allocation to affinity, $f_A$ [-]; (g-i) fractional allocation to nutrient uptake, $f_V$ [-]; nutrient limitation term of the FS variant, $L_N$ [-] (j) and fractional allocation to carbon fixation of the IA and DA variants, $f_C$ [-] (k-l); and (m-o) light saturation of photosyntehsis, $L_I$ [-] as simulated by the FS (left); IA (center) and DA (right) variants.

do not have a significant effect on the model results, as evidenced by a sensitivity experiment, where $\hat{\mu}_{\mathrm{net}}$ was constrained to positive values for the FS variant (results not shown). In comparison to the acclimative variants, $R_{\mathrm{Chl}}$ of the FS variant is smaller during the spring bloom, but larger during summer, reasons for which are explained below.

For the most part, primary production and relevant dynamics take place within roughly the upper 50m in the simulated system (Figs. 4-6). A comparison of average quantities in this zone (Fig. 7), in combination with vertical profiles throughout
the water column during different times of the year (Fig. 8), as estimated by the three model variants reveal differences between model variants that are not resolved by the contour plots (Fig. 4–6). In both the IA and DA variants, DIN concentrations are almost entirely depleted before the onset of winter mixing in early November, with minimum concentrations of $\sim$0.005 mmolN m$^{-3}$ near the surface. In the FS variant DIN remains higher (minimum concentration of $\sim$0.7 mmolN m$^{-3}$ near the surface (Fig. 7a, Fig. 8Ja,Na). $Q$ and $f_C$, as estimated by the IA and DA variants are nearly identical throughout the season
(Fig. 7b,c), but differences arise during winter. For DA, $Phy_{\mathrm{C}}$ and $Phy_{\mathrm{N}}$, hence $Q$, become vertically homogeneous due to rapid turbulent mixing (Fig. 4c,f,i, Fig. 8Fb). However under the instantaneous acclimation assumption in the IA variant, no

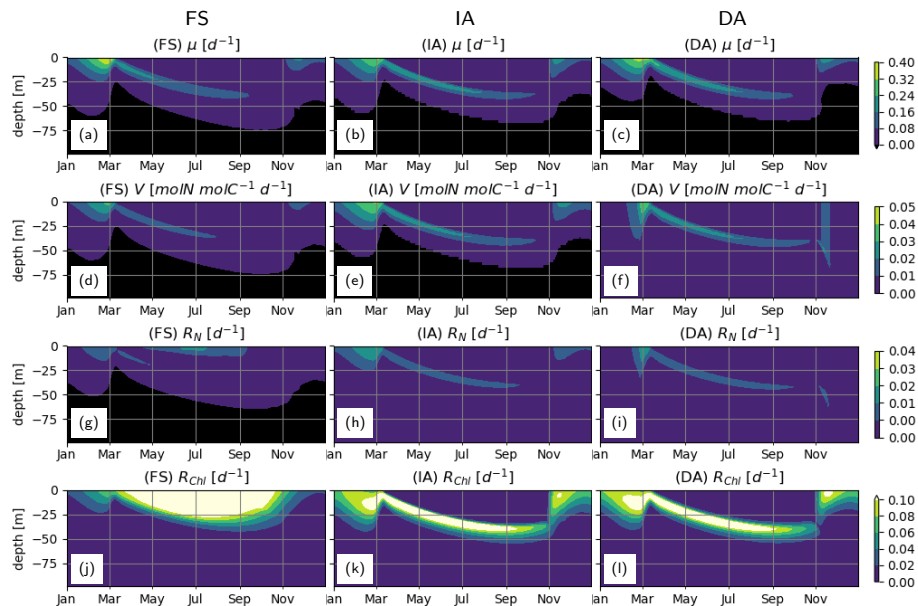

**Figure 6.** Phytoplankton growth, uptake and respiration rates. (a-c) Net growth rate, $\mu$ [d$^{-1}$], (d-f): specific uptake rate, $V$ [mmolN mmolC$^{-1}$ d$^{-1}$] and respiration costs of (g-i) N uptake, $R_N$ [d$^{-1}$] and (j-l) chlorophyll maintenance and synthesis, $R_{\text{Chl}}$ [d$^{-1}$] as simulated by the FS (left); IA (center) and DA (right) variants.

matter how well mixed the water column may be, vertical gradients persist for the optimal $Q$ values between the surface and deeper layers during winter (Fig. 8Fb,Nb).

During winter and the spring bloom in March-April, nutrient limitation is almost non-existent for the acclimative variants as

indicated by $f_C$ approaching unity (Fig. 7c), whereas for the FS variant, a degree of nutrient limitation persists (as indicated by $L_N < f_C$), owed to the saturating behavior of the Monod function to the nutrient concentrations. During late summer (July to October), nutrient limitation becomes less severe for the FS variant than for the acclimative variants in the surface layers (i.e., $L_N > f_C$, Fig. 8Jc,Oc). The relatively high $L_N$ (minimum: 0.12) of the FS variant results from the incomplete DIN depletion as simulated by the FS variant as mentioned above, and the linear response of the Monod function to substrate concentrations at

low levels (Eq. (15). In contrast, for the IA and DA variants, $Q$ approaches $Q_0$ and $f_V$ approaches its maximum value of 0.5, causing (through Eq. (11)) severe nutrient limitation, as $f_C$ approaches to zero (minimum: 0.005) near the surface.

The cellular net growth rate, $\mu$, as estimated by the FS variant is slightly faster than those of the acclimative variants during winter/spring near the surface (e.g. Fig. 8Ff,Mf) but becomes slower right after the spring bloom (e.g. Fig. 8Af), and stays low throughout the summer (Fig. 7f, Fig. 8Jf). It should be noted that, the chloroplast-specific growth rate, $\hat{\mu}$, which is maximized

for the acclimative variants through photoacclimative flexibility (Section 2.2.4), is always higher than that calculated by the FS variant, as expected (not shown). As the chloroplast specific chlorophyll maintenance and synthesis costs, $\hat{R}_{\text{Chl}}$ is scaled to the cellular level (through multiplication with $f_C$, Eq. (25)), the resulting $R_{\text{Chl}}$ for the FS becomes lower than those of the

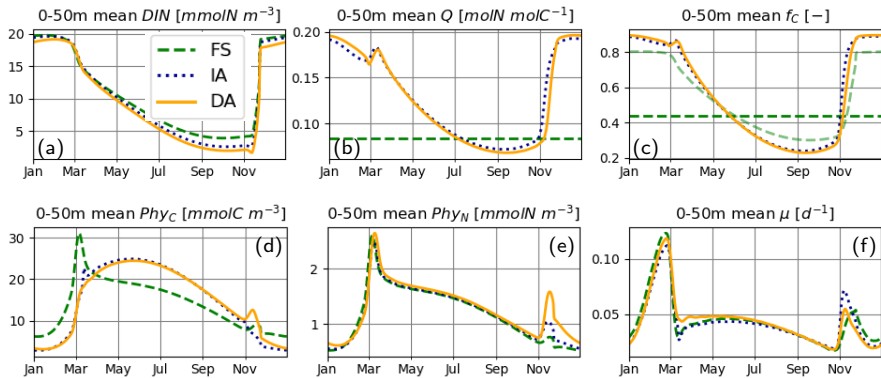

**Figure 7.** Upper 50m averages of critical variables. (a) DIN [mmolN m$^{-3}$], (b) phytoplankton $Q$ [mmolN mmolC$^{-1}$], (c) $f_C$ [-] (in addition $L_N$ [-] for FS, shown with pale broken line), and (d) $Phy_C$ [mmolC m$^{-3}$], $Phy_N$ [mmolN m$^{-3}$] (e), and $\mu$ [d$^{-1}$] (f), as simulated by the FS (dashed green line), IA (fine-dashed dark blue line) and DA (continuous orange line) variants.

acclimative variants, given that the prescribed $f_C$ of the FS variant during this time period is smaller than the dynamically calculated values by the acclimative variants (Fig. 7c; Fig. 8Fc,Mc). The lower $R_{Chl}$ of the FS variant, in turn, explains the

higher $\mu$ during the spring bloom (Fig. 7f). When the chloroplast size of the FS variant is assumed to be proportional to $L_N$ as explained in the Appendix B, estimated growth rate becomes similar to those of the acclimative variants (Fig. B2f). During summer, this effect becomes reversed: high $R_{Chl}$ as estimated by the FS variant in the surface layers (Fig. 6j vs k-l) contributes to the relatively low $\mu$ estimated by this variant (Fig. 7; Fig. 8Jf): in addition to the higher $\hat{\mu}$, the IA and DA variants achieve lower $R_{Chl}$ (Fig. 6j-l) through lower $\hat{\theta}$ (Fig. 5a-c) and $f_C$ (Fig. 7c, Fig. 8Jc) at the surface.

During the spring bloom, C bound to phytoplankton, $Phy_C$, simulated by the FS variant exceeds those of the IA and DA variants (Fig. 7d), whereas the differences between the N bound to phytoplankton, $Phy_N$ as simulated by different variants are much smaller (Fig. 7e). This discrepancy between C and N content of phytoplankton is due to the decoupling in the acclimative variants: due to the lower value of the prescribed $Q$ of the FS variant (based on the spatio-temporal average of the values simulated by IA) during winter-spring season (Fig. 7b), a larger amount of C-biomass can be synthesized per N taken up

in comparison to the acclimative variants, explaining therefore the higher $Phy_C$ simulated by the FS. The sensitivity of $Phy_C$ of the FS variant is evidenced also by a strong reduction of $Phy_C$ (in contrast to relatively unaltered $Phy_N$) during the spring bloom in response to a doubling of the prescribed $Q$ (not shown). During summer, the FS variant estimates considerably lower values of $Phy_C$ compared to the IA and DA variants (Fig. 7d) whereas the simulated $Phy_N$ concentrations remain to be similar (Fig. 7e). Therefore the higher $Phy_C$ concentrations simulated by the acclimative variants during this period are promoted by

lower $Q$ (Fig. 7b, Fig 8Jb) in the surface layers.

Differences between the IA and DA variants emerge especially right after the spring bloom and autumn destratification. After the spring bloom, growth rate simulated by the IA variant (until May) becomes lower than that by the DA variant (Fig. 7f) near the surface (Fig. 8Af). The main reason for this difference is the slightly lower $f_C$ of the IA variant during the winter-spring period (i.e., from December to May) near the surface (Fig. 8Fc,Mc,Ac) except for a short period at the peak of the bloom

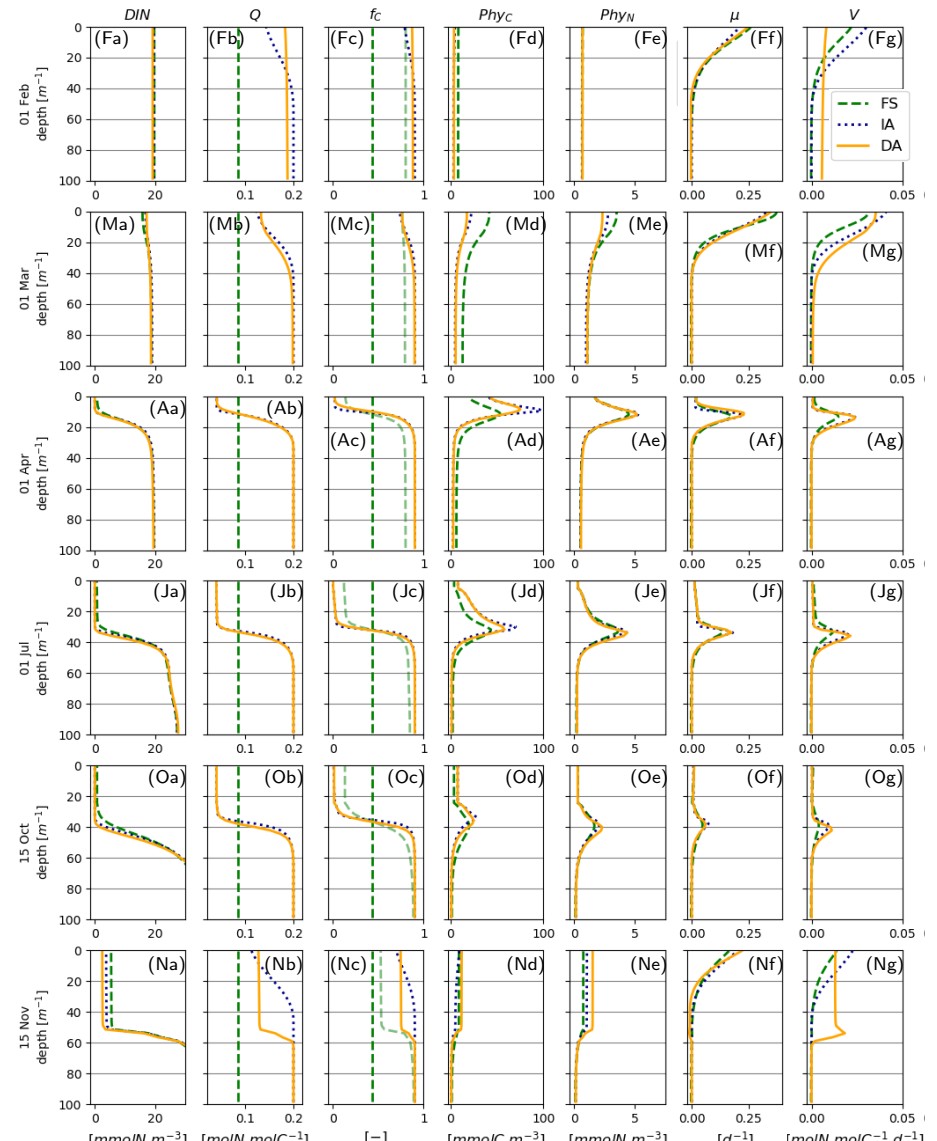

**Figure 8.** Vertical profiles on $1^{st}$ of February (indicated as F in panel label), March (M), April (A), July (J), and $15^{th}$ of October(O) and November (N) for DIN [mmolN m$^{-3}$] (indicated as a in panel label), phytoplankton $Q$ [mmolN mmolC$^{-1}$] (b), $f_C$ [-] (c), $Phy_C$ [mmolC m$^{-3}$] (d), $Phy_N$ [mmolN m$^{-3}$] (e), $\mu$ [d$^{-1}$] (f), and $V$ [molN molN$^{-1}$ d$^{-1}$] (g) as simulated by the IA (fine-dashed dark blue line), DA (continuous orange line) and the FS (dashed green line) variants, when the prescribed $\Theta$ (Table 3) is scaled with $f_C$, according to Eq. (24).

(Fig. 7c). The lower $f_C$ of IA during this period is, in turn, driven by slightly lower $Q$ (Fig. 7b, Fig. 8Mb,Ab), which also leads to slightly higher $f_V$ (see Eq. (14)). As pointed out above, the higher $Q$ simulated by the DA variant before the spring bloom near the surface is maintained by the homogenizing effect of vertical transport (which does not occur with the IA variant),

and after the spring bloom following the onset of stratification, the persistently higher $Q$ of the DA variant near the surface is reflects the lagged response captured by dynamically tracing C and N content of phytoplankton.

Following the weakening of stratification in early November (Fig. 3), a new phytoplankton bloom develops, especially as reflected by $Phy_N$ in all variants, but also by $Phy_C$ as simulated by the DA variant (Fig. 7d,e). This bloom is driven by the entrainment of DIN and phytoplankton biomass below the thermocline into the SML (compare Fig. 8Oa,d,e vs. Fig. 8Na,d,e). Under these nutrient-replenished conditions, $\mu$ is predominantly limited by light, as in winter (Fig. 8Ff), therefore monotonically increases towards the surface (Fig. 8Nf), as simulated by all variants. On the other hand, vertical distribution of $Q$ as simulated by the IA and DA variants become qualitatively different: due to the rapid turbulent mixing of $Phy_C$ and $Phy_N$ as simulated by the DA variant, $Q$ is homogeneously distributed within the SML (Fig. 8Nb), but such homogenization does not occur in the IA variant, and $Q$ is determined by the locally optimized $f_V$. Therefore, in the DA variant, a high nutrient uptake at the bottom of the SML (Fig. 8Ng), in combination with mixing within the SML can support growth near the surface (through $Q$, Fig. 8Nb), whereas in IA, growth and uptake dynamics are always coupled by definition, and determined by local physiological states only, as in the FS variant. The decoupling of (growth and uptake) rates and re-shuffling of $Q$ as simulated by the DA variant appears to allow faster uptake of nutrients in comparison to the IA variant within the SML (Fig. 8Ng). A related mechanism potentially contributing to the higher nutrient uptake rates is again a time-lag effect: in the DA variant, the nutrient-starved phytoplankton (i.e., the low $Q$, see Fig. 8Ob) in the SML corresponds to a higher nutrient demand.

C:N of detritus as estimated by the FS variant approaches a constant equilibrium value throughout the water column by the end of the first year, and remains there during the third year (Fig. 9a,d). This is as expected, and this value is simply equal to the reciprocal of the prescribed constant (N:C) quota of phytoplankton, calculated as the biomass-weighted average of the Q estimated by the IA variant (Table 3). The C:N ratio of detritus, as estimated by the IA and DA variants, increases during summer (Fig. 9b,c and e,f), driven by the lower phytoplankton quotas during summer (Fig. 4).

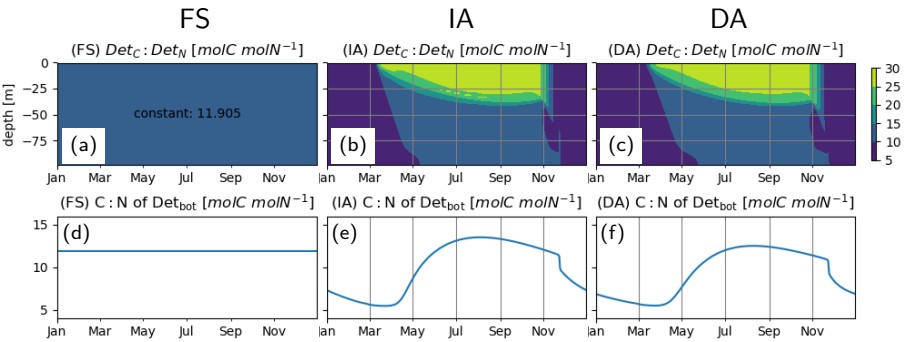

**Figure 9.** Detrital C:N [molC molN$^{-1}$] (a-c) in the entire water column, and (d-f) in the bottom layer (d-f), as simulated by the FS (left); IA (center) and DA (right) variants.

Simulated process rates determining ecosystem functioning, such as the water-column integrated Net Primary Production (NPP) and Nutrient Drawdown (NDD) rates also differ between the model variants. FS estimates higher NPP rates during

winter and the spring bloom (Fig. 10a), consistent with the higher $Phy_C$ it estimates during this period (Fig. 7d). While the NPP estimates of IA and DA are very close between the late summer (starting from September) to the spring bloom (in early March), right after the spring bloom, IA estimates suddenly decrease, as a consequence of reduced net specific growth rate, $\mu$ (Fig. 7f) as was pointed out above. Interestingly, this difference between the IA and DA is larger than the differences in $\mu$, and contrasts with the differences in $Phy_C$ averaged over the top 50m (Fig. 7d,f), but can be explained by the higher vertical covariance between the $Phy_C$ and $\mu$ in DA than in IA (Fig. 8Ad,Af). Annual average NPP rates as estimated by the FS (48.77 mmolC m$^{-2}$d$^{-1}$) and IA (45.66 mmolC m$^{-2}$d$^{-1}$) variants are respectively 8.1% and 13.9% smaller than that of the DA variant (53.06 mmolC m$^{-2}$d$^{-1}$). NDD rates (Fig. 10b) are similar during the spring bloom, but the acclimative variants become higher during summer. After the autumn mixing, NDD as simulated by the DA variant shows a spike not well reproduced by the IA and FS variants, which is driven by the fast uptake rates simulated by the DA variant throughout the SML, contrasting with those simulated by the IA variant constrained to the surface layers (Fig. 8Ng). Annual average NDD rate simulated by the DA variant (4.78 mmolN m$^{-2}$d$^{-1}$) is the highest, followed by the 8.2% lower IA (4.39 mmolN m$^{-2}$d$^{-1}$) and 14.3% lower FS (4.1 mmolN m$^{-2}$d$^{-1}$) variants.

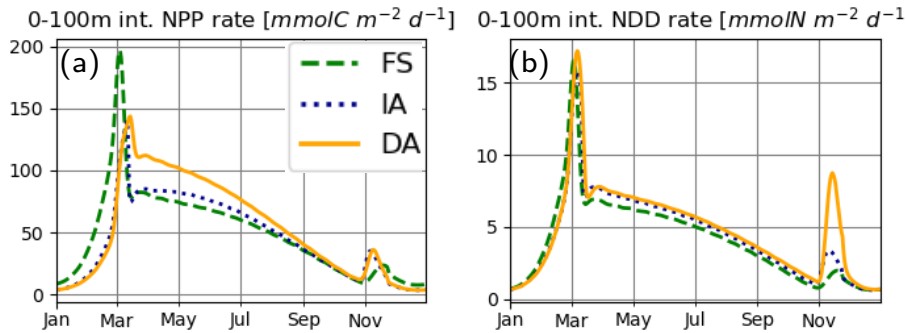

**Figure 10.** (a) Water-column integrated Net Primary Production (NPP) rate [mmolC m$^{-2}$ d$^{-1}$] and (b) water-column integrated Nitrogen Drawdown (NDD) rate [mmolN m$^{-2}$ d$^{-1}$] as simulated by the IA (fine-dashed dark blue line), DA (continuous orange line) and the FS (dashed green line) variants.

## 4 Discussion

### 4.1 Modelling variable phytoplankton composition

Elemental composition and pigment density of phytoplankton are known to vary, at both the organismal and community levels (Halsey and Jones, 2015), as demonstrated in the laboratory and under *in-situ* conditions (Moreno and Martiny, 2018). Such variations in phytoplankton and hence detrital C:nutrient ratios have implications for C and nutrient export fluxes, including the functioning of the biological carbon pump in the ocean. Notwithstanding, in many biogeochemical models coupled to GCM's, primary producers are still unrealistically represented with a constant 'Redfield' C:N:P ratio, and/or constant Chl:C

ratio (see, e.g. the models used in Laufkötter et al., 2015). More detailed 'Quota' models exist; however, these approaches are often challenged by two major limitations: *i*) dependence on formulations that lack a clear mechanistic basis, and *ii*) their requirement for additional state variables, which increase computational costs.

A concrete example of the first problem, i.e., dependence on heuristic formulations, is the down-regulation of nutrient uptake, which is needed to avoid unrealistically high nutrient quotas in a Droop scheme. Often, down-regulation is formulated as some function (linear, e.g., Grover (1991) or non-linear, e.g., Geider et al. (1998)) of 'relative quota', with reference to a prescribed maximum value. The acclimation scheme used in this study (IA and DA variants), requires no such explicit down-regulation term, nor any prescribed maximum quota value. This is because the optimization of growth, subject to the growth vs nutrient uptake trade-off (sect. 2.2.2), accomplishes this regulation by balancing the marginal benefits of investing into nutrient uptake versus photosynthesis. This RAM approach, which links various cellular functions via trade-offs, has proven successful at reproducing various Chl:C:N:P measurements obtained in laboratory experiments (e.g. Klausmeier et al., 2004; Pahlow et al., 2013; Wirtz and Kerimoglu, 2016). Furthermore, given its mechanistic basis, this approach can be expected to reproduce biological feedbacks more realistically (Flynn et al., 2015).

Earlier studies had pointed out that representation of variable in Chl:C:N ratios of phytoplankton in models resulted in better reproduction of field observations (e.g. Doney et al., 1996; Christian, 2005; Ayata et al., 2013; Chen and Smith, 2018). Consistent with those studies, implementation of the model introduced here for simulating two oligotrophic ocean sites suggested that the the portability of phytoplankton growth models are enhanced by the variable cellular composition (Anugerahanti et al., 2021). As demonstrated by these studies, 1D setups, as we also used here are ideal computational environments for examining the behavior of phytoplankton growth models: while resolving the essential features of aquatic environments, foremost the seasonally variable vertical structuring of resources and transport rates, they increase the computational costs minimally, in comparison to the 3D models. On the other hand, realistic representation of the horizontal gradients, or investigation of the effects of phytoplankton on the biogeochemical functioning at larger scales do require 3D setups. Recent applications of these models in realistic 3D setups (Kerimoglu et al., 2017; Pahlow et al., 2020) have indicated that accounting for acclimation enhances the ability of models to reproduce field observations and large scale patterns. Moreover, a consistent representation of phytoplankton composition allows identification of potential alterations in trophic transfer efficiencies as mediated by changes in food quality of prey in response to environmental change (Kerimoglu et al., 2018; Kwiatkowski et al., 2018).

Regarding the second problem, i.e., the computational costs of resolving additional state variables, Smith et al. (2016) proposed the 'Instantaneous Acclimation' approach, according to which, the elemental composition of phytoplankton vary, but instantaneously, such that tracking these variations does not require additional state variables. As in Smith et al. (2016), we considered the same specific acclimation mechanisms of (Pahlow et al., 2013), but under the assumption that the N quota adjusts to an optimal value locally, under strictly 'balanced growth' (Burmaster, 1979, see sect. 2.2.1). While at steady-state, this is a natural consequence of any 'Droop-like' model (Burmaster, 1979), assuming this behavior to hold under transient conditions is merely an approximation. Ward (2017), using a classical Droop-approach, showed that this approximation holds well under a wide range of conditions in a 0D (box) setup. Here, for the first time, we have tested this approach in an idealized 1D setup, and shown that in many respects, the IA model and the fully explicit DA variant behave similarly. Our preliminary experiments

demonstrated that, even in an environment characterized by periodic perturbations of stratification during summer, behavior of the two variants remain similar (results not shown). This is significant, considering that IA requires only 1 state variable, whereas DA requires 2 state variables. Thus, it can be concluded that IA provides improved realism over a computationally equivalent FS approach, which ignores variations in cellular composition. For simulating a few years of the dynamics of the single phytoplankton group in a 1D setup as we did here, differences in computational costs relative to the fully dynamic variant are nearly negligible, but for simulating decades/centuries or millennia in a 3D setup (e.g., as in Pahlow et al., 2020), and/or when multiple phytoplankton clones/types (e.g., with different sizes) are considered (e.g., 350 in Dutkiewicz et al., 2020), differences in computational costs can indeed be substantial.

### 4.2   Qualitative versus quantitative differences between model variants

The capacity to store nutrients is known to be an advantageous trait for phytoplankton in temporally fluctuating environments, where greater nutrient storage capacity, e.g., by larger cells, during the nutrient-replete phase provides a competitive advantage during subsequent periods of nutrient scarcity (Grover, 1991; Litchman et al., 2009). Similarly, diffusion or active movement of nutrient-rich cells from the nutrient-replete to nutrient-rich environments, e.g., from bottom towards surface layers, has been shown to favor species with greater storage capacities (Grover, 2009; Kerimoglu et al., 2012). The IA model presented in this study cannot capture this effect, since according to this approach: *i*) growth and nutrient uptake rates are always proportional (by definition of the 'balanced growth' assumption), thus, differential benefits along a space or time continuum cannot be combined through re-shuffling of physiological states; *ii*) nutrient quotas do not have inertia, hence, lagged response, as they are instantaneously adjusted to the local (light and nutrient) resource conditions; unlike in the DA variant where $Q$ is dynamically traced (by the virtue of dynamically tracing both $Phy_\mathrm{C}$ and $Phy_\mathrm{N}$). In fact, the DA variant we considered here presumably has a weaker storage capacity compared to a classical 'Droop' model, because in our acclimative approach, allocation of resources to maximize growth can be expected to suppress 'luxury consumption' (Droop, 1968) of nutrients. Finally, it should be noted that because of the differences in the formulation of the uptake in the IA (Eq. (6)) and DA (Eq. (12)) variants, and the complex inter-dependencies between the physiological states and process rates involved ($Q$, $f_\mathrm{V}$, $f_\mathrm{C}$ $\mu$, $V$, $\hat{V}$, DIN), comparison of the response of the two variants is not straightforward during such transient phases, and a fuller understanding will require further analysis and experimentation.

Some of the differences in phytoplankton growth dynamics, as simulated by the acclimative IA and DA variants and the non-acclimative FS variant, could be reconciled by tuning the parameters. For instance, the amount of phytoplankton biomass, or the extent of nutrient depletion as simulated by the FS variant can be increased by specifying higher resource affinities (e.g., lower $K_\mathrm{N}$ or higher $\alpha$, to make up for the deficiency in the formulation of light limited growth (Oschlies and Schartau, 2005). However, improvements in these specific aspects typically result in greater discrepancies in other aspects, such as the timing and magnitude of the spring bloom, or winter concentrations of nutrients and phytoplankton. In other words, in terms of model performance, trade-offs exist between multiple objectives. Such trade-offs become more obvious when attempting to simulate multiple environments characterized by different resource conditions (e.g., multiple sites, or the same site in two different time

periods) with a single parameter set (Anugerahanti et al., 2021). How acclimative flexibilities impact the sensitivity of models to parameter perturbations remains to be an open question.

The RAM approach used here, as in 'adaptive dynamics' approaches (Follows and Dutkiewicz, 2011), ambiguously reflects processes at multiple organismal scales. For instance, higher $f_A$ and $f_V$ and lower $\hat{\theta}$ at the surface layers during summer (Fig. 5), which agrees with lower light harvesting and higher nutrient harvesting investment as found by Bruggeman and Kooijman (2007), can be attributed to: *i*) evolutionary adaptation of new species (which would be more relevant in a longer-term simulation), *ii*) selection among existing species that had been pre-adapted to these conditions, and *iii*) individual-level acclimation. Optimality-based acclimative models can thus capture some key community-level effects of evolutionary and ecological dynamics, without explicitly resolving competing species or groups (Smith et al., 2011). The same idea underlies the recent work of Chakraborty et al. (2020), where they described the changes in community composition by assuming that the trophic strategy of the entire plankton community is optimized instantaneously.

Some features, such as the dense and thin chlorophyll layers at the thermocline as captured by the acclimative variants (Fig. 4) seem qualitatively irreproducible by the FS variant even for a single site and time period. This is because multiple dependencies are necessary for capturing this feature, namely the unimodal distribution of chlorophyll density over depth (Fig. 4) and the steep increase in chloroplast size with depth near the thermocline (Fig. 5), as well as the thermocline being the best compromise between light- and nutrient- limitation (Fig.5). The FS variant includes only the last dependency, because it lacks acclimation, and is therefore unable to produce such thin chlorophyll layers. When the chloroplast size is assumed to vary, and diagnosed by the nutrient limitation term, such that the vertical Chl:C increases monotonically with depth, the vertical distribution of Chl can be partially captured (Appendix B).

We also found differences in system-level metrics such as NPP and NDD (e.g., Bergeron and Tremblay, 2014; Johnson et al., 2017) rates as simulated by different variants. For both metrics, DA estimates were about 10% higher than the FS and IA variants, with FS estimates systematically skewed towards earlier in the season. It should be noted that, for the FS variant, prescribed $Q$, which, in this study was based on the results of the IA variant, but normally is effectively a free parameter (although the common approach is to set it to the Redfield proportions), largely determines the estimated $Phy_C$, and related quantities, such as NPP rates. For instance, doubling the $Q$ of FS results in only a few percent further underestimation (relative to DA) of the NDD rate (annual average: 3.89, instead of the original 4.1 $\mathrm{mmolN\,m^{-2}d^{-1}}$, which corresponds to 18.6% lower than the DA estimate, instead of the original 14.3%), whereas it leads to more than 50% lower estimates of NPP rate (23.26 $\mathrm{mmolC\,m^{-2}d^{-1}}$) in comparison to that of the DA variant. Some FS variants are not based on C, and not N as in this study (i.e., the explicit state variable is the C bound to phytoplankton). For those models, instead of the NPP, NDD rates may be more sensitive to prescribed $Q$. In contrast to the FS variant, with the IA variant, the total C and N content, and growth and nutrient uptake rates of the phytoplankton, thus the system-level process rates, like the PPR and NDD rates are determined by the same set of parameters governing the fully explicit DA variant.

## 4.3 Physiological Flexibility and Environmental Feedbacks

The well known links between the composition of phytoplankton and the biogeochemistry of their ambient environments imply feedbacks, which are important in ecology, environmental science, and water quality studies. These feedbacks can be mediated by both physiological acclimation and evolutionary adaptation (Moreno and Martiny, 2018), with the latter typically understood to operate on much longer timescales. However, acclimation and adaptation do interact in eco-evolutionary dynamics, and for plankton they may even occur on similar timescales (Smith et al., 2011; Edelaar and Bolnick, 2019). Disentangling their effects is challenging, and debate continues as to the relative roles of acclimation and evolutionary adaptation in determining the observed patterns of variation. For example, although Sharoni and Halevy (2020) attribute observed seasonal variations in the elemental composition of detritus to seasonal sorting among various well adapted species, that conclusion was based on the assumption that acclimation implies a lack of nutrient limitation, which is not the assumption underlying most acclimative models, including ours. For example, the near-zero values of $f_C$ in the upper 25 m during summer months (Fig. 5k,l) indicate extreme nutrient limitation, which prevents growth in the surface layers (Fig. 6b,c). In any case, only models that account for the relevant flexibilities and variations in the composition of phytoplankton can be expected capture such feedbacks in a general, yet realistic sense, which is necessary to correctly assess the relative roles of plankton-related processes in biogeochemical cycles.

An important link between flexibility and environmental feedbacks is the role of phytoplankton in determining the elemental composition of particulate matter (Redfield, 1934). Key mechanisms involve the activities of nitrogen fixers and denitrifiers (Redfield, 1958). However, given the differences in stoichiometry of macromolecules involved in various cellular functions (Geider and La Roche, 2002), a consistent description of the acclimation of phytoplankton is necessary to represent realistically the variabilities in elemental composition of particulate matter, hence, export fluxes. Fixed stoichiometry models erroneously predict constant elemental composition of detrital matter production, as demonstrated by our FS variant in this study. The so called 'Droop' models have been shown to capture the observed seasonal increase in detrital C:N ratios during summer, reflecting nutrient limitation of phytoplankton (e.g., Mongin et al., 2003). Representing the growth and uptake terms consistently using the RAM framework, the DA variant resolves the seasonal and vertical variations in the elemental composition of particulate matter (Fig. 9). With some exceptions, estimates of the IA variant are nearly identical to those of the DA variant, thereby implying that a more realistic representation of these can be achieved at no additional computational cost compared to a fixed-stoichiometry models.

## 4.4 Present implementation, challenges and perspectives

Moving a coupled hydrodynamic-biogeochemical models from a 0-D setup to a spatially explicit setup can be error-prone and time consuming. The Framework for Aquatic Biogeochemical Models, (FABM Bruggeman and Bolding, 2014), provides an easy to use coupling layer that connects a hydrodynamic model with multiple biogeochemical sub-models. FABM specifies how the these models communicate by separating the hydrodynamics and biogeochemical models, with FABM acting as a glue layer in between. The biogeochemical model in this framework operates locally in space where the local source and sink terms

are computed based on the local state and environment, making it feasible to scale up from 0-D to n-D, and swap different hydrodynamic models. FABM also provides mechanisms to pass other environmental data, such as temperature, salinity, and pH, from different submodules, as long as the biogeochemical models register any dependencies during initialisation. Therefore, complex description of the biogeochemical models can be partitioned into several sub-modules. The modular implementation of our model in FABM, specifically, the isolation of the phytoplankton module (Fig. 1) is expected to facilitate studies with multiple phytoplankton types. For example, without changing the model code nor recompiling, just through changing a configuration file, it is possible to include further types (see Bruggeman and Bolding, 2014), which can be parameterized, e.g., according to cell size (as in, e.g. Smith et al., 2016; Dutkiewicz et al., 2020). Moreover, the isolated phytoplankton module can be relatively easily coupled with or incorporated into existing models, especially those implemented in FABM.

Currently, the model simplistically accounts for the grazing losses to higher trophic levels with a quadratic mortality term (Table 1), without describing explicitly the dynamics of preditors. This limitation may prohibit realistic applications to highly productive ecosystems, where the strength of top-down control exhibits strong seasonality (e.g., Maar et al., 2014; Sailley et al., 2015). However, this problem can be easily resolved by adapting an existing zooplankton module available for FABM, such as the N-only resolving module in the 'NPZD' example provided in the standard FABM library (Bruggeman and Bolding, 2014). An explicit consideration of zooplankton can expected to introduce additional complexities: depending on how zooplankton C and N co-limitation is described, variabilities in phytoplankton stoichiometry may affect zooplankton growth (e.g., Mitra et al., 2007; Branco et al., 2018; Kerimoglu et al., 2018) and in turn, depending on the parameterization of zooplankton excretion and remineralization processes, subsequent phytoplankton blooms may occur. While it was our explicit aim to avoid such complicated indirect effects and focus on the direct effects of acclimation mechanisms on phytoplankton growth in this study, coupling the presented model to a larger ecosystem model including herbivores and their predators would allow investigating the propagation of these effects throughout the food web in a cost-effective manner.

For simplicity, we have traced only N here fully (e.g., no explicit DIC, but only DIN, see Eq. (4)) and the model is therefore conservative with respect to N, but not with respect to C. When multiple nutrient elements in the dissolved inorganic material pool (e.g., C, N, and P) are resolved, maintaining mass balance becomes more complicated under the IA assumption (see Smith et al., 2016; Ward, 2017). FABM-implementation of a carbon-based version of the model that resolves the C and N cycles is being currently developed, which are we are planing to present in a separate study. The extended model will be able to resolve C, N, P and micronutrient cycles based on a common mass-balance formalism, and therefore allow investigating the validity of assuming instantaneous optimization of C:N:P:micronutrient ratios under various environmental conditions (relevantly, see Bonachela et al., 2013). However, for various ecological applications, especially those resolving multiple phytoplankton types, tracing only one nutrient element, as in the current study may be sufficient and more convenient.

In the current study, we focused on the differences between the fully acclimative IA and DA variants, and an entirely non-acclimative variant. Our acclimation scheme consists of four acclimative flexibilities: variability of internal nutrient quota, optimization of uptake vs growth trade-off, optimization of maximum uptake vs. affinity trade-off, and optimization of chlorophyll density in chloroplast density (and as an additional half-step, size of the chloroplast, see Appendix B). In a future study, we are planning to investigate the relative importance of each of these flexibilities for the organismal fitness under various

environmental conditions: such an assessment would not only help the model developers to prioritize the research needs, but
may also provide insights into the evolution of these acclimative flexibilities.

## 5 Conclusions

In this study, we present a FABM-implementation of the 'NflexPD' model, and the behavior of three variants it can emulate: a
Fixed Stoichiometry (FS) variant that lacks any acclimative flexibility and explicitly tracks only N bound to phytoplankton; a
Dynamic Acclimation (DA) variant that resolves various acclimative flexibilities by explicitly tracking the C and N in phyto-
plankton; and the Instantaneous Acclimation (IA) variant that resolves the same flexibilities as the DA variant, but by tracking
the N in phytoplankton as in the FS variant.

By applying the NflexPD model coupled to an idealized, 1D water column model, we aimed to understand: i) whether and
how the behavior of the IA and DA variants differ; and ii) whether and how the behavior of the acclimative variants differ
from the non-acclimative, fixed stoichiometry variant. With regard to the first of our objectives, we found that behavior of IA
is stable and in many respects very similar to that of DA, although differences arise during the spring and autumn transitions,
owing to the lagged response and vertical transport of nutrient quotas in the DA variant. With this, our study provides proof-
of-concept that the IA approach is applicable in spatially-explicit setups, and hints at conditions under which deviations from
the fully explicit variant can be expected. With regard to the second objective, we found substantial differences between the
behavior of the FS and acclimative variants: with the particular parameterization we show-cased here, the acclimative variants
estimated smaller spring blooms, but sustained growth during summer and stronger nitrogen depletion in the surface layers,
as well as steeper chlorophyll layers at the thermocline; and unlike the FS variant, they can reproduce the variabilities in
C:N of particulate matter. Moreover, a subset of quantities estimated by the FS variant, such as the phytoplankton biomass
and NPP rates were found to be strongly sensitive to the prescribed parameters such as $Q$, which, in this study was derived
as a spatio-temporal average from the IA variant, but is typically an adjustable parameter, implying thus a higher degree of
freedom. These qualitative differences provide insight into the impact of acclimative flexibilities on model response, and their
ecosystem-scale implications. The model implementation presented here tracks only N as dissolved nutrient, which restricts
its utility in biogeochemical studies that require a complete representation of the cycling of multiple elements, but it can be
readily used in various ecological contexts.

*Code availability.* For running the model and reproducing the results presented in this study, FABM and GOTM need to be downloaded
and installed. See https://github.com/fabm-model/fabm/wiki/GOTM for the instructions. The version of the FABM-NflexPD used in this
manuscript has been stored in Zenodo repository https://doi.org/10.5281/zenodo.4761937. Instructions for compiling FABM-NflexPD for
GOTM-FABM and a 0D setup are provided in README.md. The 'src' folder contains the Fortran codes. The model was implemented
as two separate modules: the 'phy.F90' module that describes phytoplankton growth and the 'abio.F90' module that describes everything
other than phytoplankton (Fig. 1). The phytoplankton module can reproduce the behavior of all three different variants considered in the

 manuscript through optional parameters. The 'testcases' folder contains the configuration (yaml) file that was used to produce the results presented in this manuscript, thereby providing examples of how each variant can be initiated.

## Appendix A: Details of Derivations

### A1  $R_{\mathrm{N}}$ for FS variant

According to Eq. (7), $R_{\mathrm{N}} = \zeta_{\mathrm{N}} \cdot V$. For the DA and IA variants, $V$ can be calculated externally (Eq. (12)), hence so can be $R_{\mathrm{N}}$. For the FS variant on the other hand, there is no explicit solution for $V$, but it can only be calculated as a function of $\mu$, ($V = \mu \cdot Q$, Eq.(6)), and since $\mu$ in turn, depends on $R_{\mathrm{N}}$ ($\mu = \mu_{\mathrm{net}} - R_{\mathrm{N}}$, Eq. (7)), $R_{\mathrm{N}}$ cannot be directly calculated. Expanding the terms in Eq. (7) according to Eqs. (6), (13) and (20):

$$\mu = \hat{\mu}_{\mathrm{net}} \cdot L_{\mathrm{N}} - \zeta_{\mathrm{N}} \cdot \mu \cdot Q \tag{A1}$$

reorganizing:

$$\mu = \frac{\hat{\mu}_{\mathrm{net}} \cdot L_{\mathrm{N}}}{1 + \zeta_{\mathrm{N}} \cdot Q} \tag{A2}$$

substituting this with $\mu$ in:

$$R_{\mathrm{N}} = \zeta_{\mathrm{N}} \cdot V = \zeta_{\mathrm{N}} \cdot \mu \cdot Q \tag{A3}$$

we obtain a $V$-independent expression for $R_N$:

$$R_{\mathrm{N}} = \zeta_{\mathrm{N}} \cdot \frac{\hat{\mu}_{\mathrm{net}} \cdot L_{\mathrm{N}}}{1 + \zeta_{\mathrm{N}} \cdot Q} \cdot Q \tag{A4}$$

It can be verified that, when this term is substituted in $\mu = \mu_{\mathrm{net}} - R_N$, it yields $\mu = \mu_{\mathrm{net}} - \zeta_{\mathrm{N}} \cdot \mu \cdot Q = \mu_{\mathrm{net}} - \zeta_{\mathrm{N}} \cdot V$, i.e., Eq. (7), implying that using $R_{\mathrm{N}}$ in Eq. (A4) for the FS variant makes Eq. (7) valid for the FS variant as well.

### A2  Optimal $Q$ and $f_{\mathrm{V}}$

In Eq. (7), substituting $\mu_g$, $R_N$ and and $R_{Chl}$ with the expanded forms in Eqs. (13), (20), and (25), respectively, and subsequently expanding $\theta$, using Eq. (24)):

$$\mu = f_{\mathrm{C}} \hat{\mu}_g - \zeta_N f_{\mathrm{V}} \hat{V} - (\hat{\mu}_g + R_M^{Chl}) \zeta_{Chl} \hat{\theta} f_{\mathrm{C}} \tag{A5}$$

Reorganizing:

$$\mu = f_{\mathrm{C}} \left[ \hat{\mu}_g (1 - \zeta_{Chl} \hat{\theta}) - \zeta_{Chl} \hat{\theta} R_M^{Chl} \right] - \zeta_N f_{\mathrm{V}} \hat{V} \tag{A6}$$

Substituting the term in square brackets with $\hat{\mu}_{\text{net}}$ based on Eq. (7) and expanding $f_C$ using Eq. (11)):

$$\mu = \left(1 - \frac{Q_0}{2Q} - f_V\right)\hat{\mu}_{\text{net}} - \zeta_N f_V \hat{V} \tag{A7}$$

At this point, it can be readily recognized that Eq. (A7)) is equivalent to the Eq. (5) in Pahlow and Oschlies (2013), only difference being their denotion of $\hat{\mu}_{\text{net}}$ as $\hat{\mu}^I$. Note that their formulation of respiration losses within the chloroplast as a fraction of gross growth with respect to chloroplast (i.e., $\hat{\mu}^I = \hat{\mu}_g^I(1 - \zeta^C)$ in their notation), differs from the more precise formulation we used here, that considers a base loss rate independent of gross growth. However, considering that $\hat{\mu}_{\text{net}}$ (just like their $\hat{\mu}^I$) is independent of $Q$ and $f_V$, the solutions provided by Pahlow and Oschlies (2013) for $f_V^o$ (i.e., their Eq. (9), our Eq. (14)) and $Q$ (their Eq. (10), our Eq. (10)) can be directly used, only after replacing $\hat{\mu}^I$ in the original solutions with $\hat{\mu}_{\text{net}}$ for the latter.

## Appendix B: FS variant with a variable chloroplast size

Given the similar roles of $f_C$ in the IA and DA variants and the nutrient limitation term, $L_N$, in the FS variant for calculating $\mu_g$ (see Section 2.2.2), $L_N$ can be considered as a proxy for the relative size of the chlorplast. Therefore, $f_C$ in Eq. (24) and (25) can be replaced by $L_N$ for scaling the chloroplast-specific chlorophyll density and respiration costs in order to represent spatio-temporal variations of the cellular Chl:C ratio and proportional respiration costs.

When this is done, unlike the original results shown in the main text (Fig. 4m), a spatio-temporally variable Chl:C ratio (Fig. B1c) is obtained. Monotonically increasing $L_N$ with depth during summer (Fig. 5j) reduces Chl at the surface, and enhances it at the deeper layers relative to the Chl pattern obtained with constant Chl:C (compare Fig. 4m vs. Fig. B1a). However, due to the missing unimodal signal through $\hat{\theta}$ as accounted for by the IA and DA variants (see Fig. 5b,c), the resulting Chl pattern is still qualitatively different from those estimated by the truly acclimative variants (compare Fig. B1a vs. Fig. 4k,l). Furthermore, the relatively higher value of $L_N$ during the spring bloom under nutrient-rich conditions (Fig. 5j) relative to the prescribed, constant value of $f_C$=0.44 used for the case with constant chloroplast size (hence, constant Chl:C) shown in the main text as yielded by the prescribed values of $f_V$, $Q$ and $Q_0$ (Table 3 and Eq. (11)), results in greater $R_{\text{Chl}}$ (compare Fig. B1d vs Fig. 6j). Hence, net cellular growth rate, $\mu$ becomes slightly lower than in the constant chloroplast case during the spring bloom (compare Fig. B1b vs Fig. 6a). On the other hand, during summer, relatively lower values of $L_N$ make $R_{\text{Chl}}$ lower, and $\mu$ greater compared to the constant chloroplast case.

Dynamics of the $Phy_C$ within the top 50m as simulated with this flavor of the FS variant with variable chloroplast size are almost identical to those simulated by the standard, 'vanilla' version with constant chloroplast size (compare Fig. B2d with Fig. 7d). Relatively higher $R_{\text{Chl}}$ at nutrient-rich conditions during winter and early spring makes the winter $Phy_C$ concentrations (Fig. B2d) lower in comparison to the standard case (Fig. 7d). On the other hand, relatively lower $R_{\text{Chl}}$ at nutrient-scarce summer conditions make the $Phy_C$ concentrations (Fig. B2d) slightly higher than the standard case (Fig. 7d). As a result, the the average DIN concentrations in the surface 50m become slightly lower than the standard case (Fig. B2a vs. Fig. 7a), which is, better observed in lower $L_N$ (Fig. B2c vs. Fig. 7c), due to the strong response of the function at low concentrations.

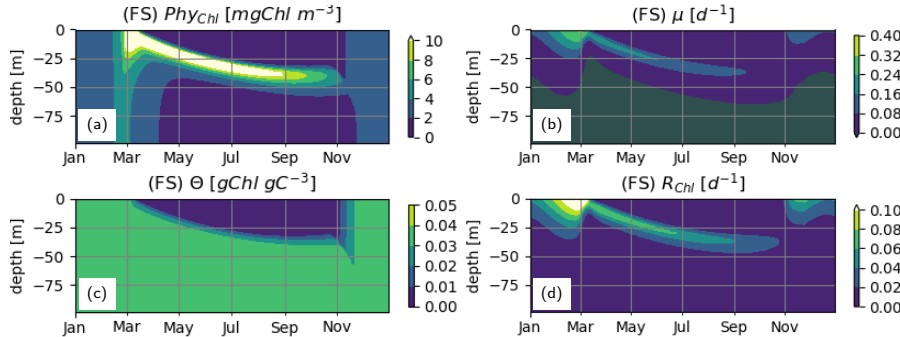

**Figure B1.** Phytoplankton (a) Chl concentration, $Phy_{Chl}$ [$mgC\ m^{-3}$]; (b) net growth rate, $\mu$ [$d^{-1}$]; (c) Chl:C, $\Theta$ [$gChl\ gC^{-1}$]; (d) respiration cost of chlorophyll maintenance and synthesis, $R_{Chl}$ [$d^{-1}$] as simulated by the FS variant, when the prescrebed $\Theta$ (Table 3) is scaled with $f_C$, according to Eq. (24).

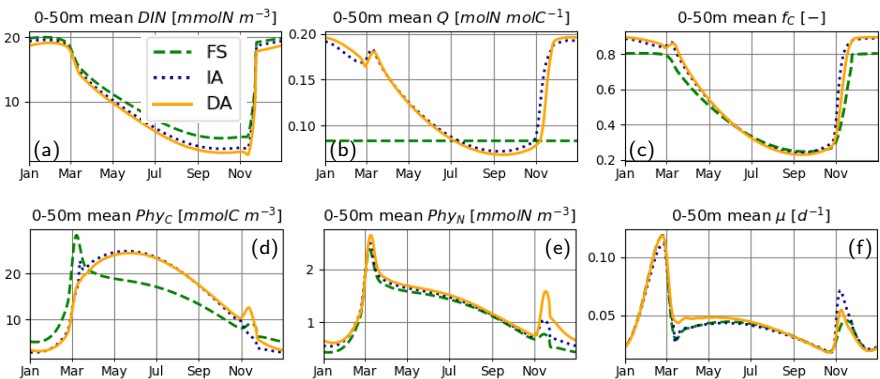

**Figure B2.** Like Fig. 7, but when for the FS variant, prescribed $\hat{\theta}$ (Table 3) is scaled with $L_N$, i.e., replacing $f_C$ with $L_N$ in Eq. (24).

Despite the differences in details explained above, especially based on the preserved qualitative differences in simulated $Phy_C$ concentrations between the FS and acclimative variants, it can be concluded that the overall conclusions are insensitive to the assumption regarding the size of the chloroplast of the FS variant.

*Author contributions.* OK and SLS designed the study, OK implemented the model code, performed the model runs and prepared the figures, PA tested the model, all authors contributed to the writing of the manuscript.

*Competing interests.* No competing interests are present.

*Acknowledgements.* We are grateful to Markus Pahlow for critically reviewing the model description, helping to correct the implementation of IA and DA variants, and providing the code for the model diagram. OK, SLS and PA were supported through a bilateral research project, funded jointly by the German Research Foundation DFG (grant no. KE 1970/2-1, P.I.: OK) and the Japan Society for the Promotion of Science, JSPS (P.I.: SLS). At an early stage, OK was additionally supported by the Federal Agency of Environment (Bundesumweltamt, UBA; Grant no. 3718252110, P.I.: OK). We acknowledge the developers of the open-source software used in this study, foremost FABM and
GOTM. We are grateful for the comments of two anonymous reviewers, which helped significantly improving the manuscript.

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
