# Peer review of "FABM-NflexPD 1.0: Assessing an Instantaneous Acclimation Approach for Modelling Phytoplankton Growth"

_Geoscientific Model Development, 2020_

## Author Comment (AC3)

**Response to Reviewer #1**

We thank the reviewer for their careful reading of the manuscript, and constructive suggestions. Below, the original comments of the reviewer are quoted verbatim, and our responses are provided right after each comment.

*"General comments*
*The manuscript describes three variants (FS, IA, and DA) in a biogeochemical model (FABM-NflexPD) coupled with a hydrodynamical model (GOTM). It clearly describes the differences between a fixed stoichiometry (FS) to more complex variants considering a dynamic acclimation (DA) and an instantaneous acclimation (IA).*
*Based on an idealized set up, the response of each model variant to irradiance and temperature is studied. Results show that adding higher complexity to the model creates differences in the model output that cannot be negligible. It is an interesting manuscript related to the scope of the journal and model development.*
*Overall, this manuscript presents an interesting approach to understand the dynamics of adding a flexible stoichiometry. However, the authors pose two objectives for this manuscript that are not fully addressed, so it is suggested that more discussion is added to fully address the objectives specified. It is also necessary that the authors specify why they used an NPD model instead of the NPZD model available from the FABM library. Moreover, I think the manuscript is lacking a more extensive discussion (or conclusions) about the novelty of this work, which could be done by, for example, extending the discussion (or conclusions) to explain why this work is relevant, given the model development is interesting but there is not a comparison to observations. Under the idealized set up of the different model variants created in this work and without comparison to observations, the authors should clearly state the applicability of their work and why it is important to have an IA variant considered in further biogeochemical modeling research.*
*This manuscript is within the standards of excellence of the journal, but the authors should address the comments suggested here."*
Response: we will extend the discussion to better communicate how the objectives are met, novelty of the work, applicability of the model and reasons to consider acclimation in biogeochemical modeling research. Regarding not having used an NPZD scheme, but an NPD scheme: indirect effects of zooplankton make it difficult to understand the direct effects of including acclimative mechanisms, please see our detailed response below to the comment for L450-454 in the Discussion section.

*"Specific comments*
*Introduction*
*L29 It is mentioned that acclimation models are now commonly used, but only Geider's model is included. Could it be added a couple more examples apart from Geider's paper?"*
Response: We will mention a few additional examples. We also realized that the example given for constant N:C and variable Chl:C ratio should not be Moore et al., (2002), where N:C was variable, but Moore et al. (2004), we will correct this.

*"L34-35 In the sentence 'Models that account for variations in cellular composition are indeed more likely to provide more realistic estimates of phytoplankton biomass and biogeochemical fluxes'. Can it be briefly explained how are estimates provided more realistic based on the variable cellular ratios?"*
Response: we will explain that the estimates of the models with variable cellular composition become more realistic specifically with respect to comparison against in situ and satellite observations and biosynthesis, and consequently the drawdown of nutrients.

*"L38 clones/types – what is the meaning of this? Could it be specified what are clones/types?"*
 Response: we will specify that these are 'phytoplankton functional types or clones'.

*"L42 ... 'response to changes in resource environment' ... - please describe what those resources are."*
Response: we will specify the resources as mineral nutrients and light.

*"L48-54 Please add the units of Q, fv, fA, and the Chl:C."*
Response: we will add units for $Q$ [molN/molC] and $θ$ (gChl/molC) and specify that $f_V$ and $f_A$ are dimensionless.

*"L59-60 What is short-term in this case? Please specify."*
Response: we will specify that here short-term refers to hours to days.

*"L84-85 'Compared to the FS variant, do the results differ sufficiently to justify the additional complexities introduced by the IA variant?' - I think this is an interesting question to set as an objective of this study. However, it is not fully addressed in this work (see further comments in the conclusion section). If the FS model provides significantly different results than the IA model, is that a justification to add further complexity to models? Under an idealised setup and with no comparison to observations, how can you tell that the differences found are answering to your question? Because of the obvious differences between FS and IA stated in section 1.4, it is expect that the results of the simulation will differ between both approaches. I suggest this question to be rephrased, as it is hard to justify IA over FS when there's no comparison to observations, which would help the manuscript better ground the differences found. Also, add more about the answer to your objective (ii) in the discussion and conclusion section (see below comments for those sections)."*
Response: thanks for this careful observation. Although earlier work have provided ample evidence for the improved realism of variable cellular composition models as explained in the previous paragraphs of our introduction, without any comparison against observation data, we indeed cannot defend in the current study that the performance of the  IA variant is better than the FS variant, which the objective (ii) itself implies presumptively by '.. to justify the additional complexities introduced by the IA variant'. In the revised manuscript, we simplified this objective as: 'compared to the FS variant, do the results of the IA variant differ substantially?'. We will also mention that while answering these questions, we aimed at gaining mechanistic understanding of the dynamics driving the difference between the model estimates. Please see below for our response to the specific comments for the discussion and conclusion sections.

*"Model description*

*L128 Which mechanisms mentioned in the Introduction is this referring to? I am not sure to what section of the introduction to look. Could it be specified which section? Or mention those mechanisms in L128."*

Response: We are referring to the section 1.2 in the introduction. We will refer to this section and mention these mechanisms explicitly.

*"L142, eq. 10 – Please describe what Qo is and reference where it is defined (Table 3)."*

Response: we will describe Qo and other intermediate terms appearing in the equation, referring to tables where they are defined.

*"L177 – Please describe what nutrient affinity is in this section."*

Response: We will introduce the definitions for nutrient affinity, and what affinity and maximum nutrient uptake rate represent.

*"L203 – Add reference at the end of the sentence (Table 3)."*

Response: will do.

*"L248 – The sinking rate value was based on observations? Please say where was that value obtained from."*

Response: we will indicate here that the value is chosen arbitrarily for this idealized setup to induce a downward vertical flux, and that in reality the sinking rates of detritus depends on the average size and density of detritus particles being modelled and displays a vast range (Guidi et al., 2008).

*"L254 – Mention the other meteorological variables assumed as constant."*

Response: we will mention that the constant meteorological variables are wind speed, air pressure, humidity, cloud cover.

*"Results*

*L287 – can the authors extend the explanation about the effect of DIN depletion? Are the DIN depletion differences the only reason for the differences between Ln (FS) and fc (IA + DA)?"*

Response: in fact, differences between the $L_N$ (FS) and $f_C$ of the acclimative variants cannot be reduced to the differences in DIN because of the structural differences between how these quantities are computed, although they both serve in representing nutrient limitation of growth, thus they are functionally equivalent. But the incomplete depletion of DIN as simulated by the FS variant, and the high sensitivity of the Monod function (Eq.15) to the nutrients at low concentrations are the reasons for the $L_N$ values being not too low. This contrasts with the $f_C$ of IA and DA being close to zero, owed to $f_V$ being close to its maximum value of 0.5 and $Q$ being close to the subsistence quota, $Q_0$ (see Eq.11). We will extend our explanation accordingly (please see also our response to the comment for L312-313).

*"L291 – In the contour plot for the net growth rate, it looks as if FS has higher values during the spring bloom in comparison to IA and DA."*

Response: thanks for noticing this, our statement does not apply for growth rate, but for nutrient uptake rate. We will reformulate this sentence accordingly. However FS attaining higher growth rates during the spring bloom indeed deserves some explanation: this is mainly caused by the lower whole cell chlorophyll respiration cost, $R_{Chl}$ of the FS variant than those of the IA and DA variants in the surface layers during winter/spring. Observing that $R_{Chl}$ is scaled from the chloroplast-specific $\hat{R}_{Chl}$ based on $f_C$ (Eq. 25), the lower $R_{Chl}$ is a result of fixed $f_C$ of the FS variant (computed based on prescribed $f_V$ and $Q$) being lower than the $f_C$ of the acclimative variants (computed based on variable $f_V$ and $Q$) in the surface layers during spring (Fig. R1.1a, where the $f_C$ and $L_N$ of the FS variants are both shown, whereas in the manuscript, only the $L_N$ of the FS variant was shown. In the revised manuscript, we will use the version shown here). It should be noted that, when the chloroplast size of the FS variant is assumed to be variable like the IA and FS variants, but based on the nutrient limitation term, $L_N$ as described in Appendix B (Fig.R1.1b, note that now $f_C=L_N$), $R_{Chl}$ becomes higher at the surface layers, and as a result, net growth rate, $\mu$ shrinks, and is not any longer higher than those of the IA and DA variants, as shown below in Fig.R1.1c,d). We will explain these in the revised manuscript, and extend Figs. 7 and B2 with the panels c-d in Fig.R1.1, respectively

[Figure]

Figure R1.1. Upper 50 m average (a-b) $f_C$ [-] (and $L_N$ [-] for FS) and (c-d) $\mu$ [d-1], as simulated by IA (fine-dashed dark blue line), DA (continuous orange line), (a, c) default FS (dashed green line; in panel a, $L_N$ is shown with pale dashed green line. Note that in the Fig.7c of the original version of the manuscript, the prescribed $f_C$ was not shown, and only $L_N$ was shown) and (b, d) FS with variable chloroplast size (as described in Appendix B).

*"L300 – Which sensitivity experiment?"*

Response: these results were not shown, we will indicate this at the end of the sentence.

*"L304-305 – saying 'before the onset of winter mixing' is vague. Do the authors mean one day before, the week before? I suggest it is specified which day/date this is. Moreover, it is confusing to see how 'substantially' higher are the values of DIN for the FS variant in comparison to IA and DA variants. To avoid confusion, please also specify what the DIN values are for the FS, IA, and DA variants before (insert date here) the onset of winter mixing. I can see there are differences during the summer period for DIN, but is that a substantial difference? Beware of the adjectives used if there is not a quantitative comparison."*

Response: we will mention `early November' for the onset of winter mixing and refer to Fig.7a. In the parenthesis L305, we will specify the exact DIN concentrations estimated by each variant on the date the minimum concentrations are reached). Relevantly, recognizing the ambiguity in references to both times of the year and quantities in the contour plots, we will include an additional figure that shows the vertical profiles at different times of the year (exemplified for a summer profile in Fig. R1.2.

[Figure]

Figure R1.2. Summer (as represented by August, 1$^{st}$) vertical distribution of (a) DIN [mmolN m$^{-3}$], (b) phytoplankton $Q$ [molN molC$^{-1}$], (c) resources available for carbon fixation, $f_C$ [-] (and $L_N$ [-] for FS, shown with the pale green dashed-line), (d) $Phy_C$ [mmolC m−3], (e) $Phy_N$ [mmolN m$^{-3}$], and (f) $\mu$ [d$^{-1}$], as simulated by the FS (dashed green line), IA (fine-dashed dark blue line) and DA (continuous orange line) variants.

*"L312-313 – the differences between Ln (FS) and fc (IA, DA) are large to be only explained by small differences in the DIN during the summer period. Could the authors give a thought to that difference and explain it in a short sentence?"*

Response: Differences in DIN at the near-surface are stronger (e.g., minimum concentrations reached by FS and IA/DA variants are 0.005 vs 0.7 mmolN m$^{-3}$, respectively) than suggested by the top 50m averages shown in Fig.7c. Besides, as explained above in response to the comment on L287, the Monod function is strongly sensitive to the DIN at low concentrations. We will mention these as an extended explanation. We also realized that the last sentences in this paragraph can be potentially confusing therefore we will remove them.

*"L316 – Please describe more the differences between FS and IA, DA during the spring bloom. FS reaches a higher PhyC in March than IA and DA."*

Response:  We agree that the differences between the FS and acclimative variants during spring and summer need to be better explained. Although the net growth rate estimated by FS variant becomes slightly higher those estimated by the acclimative variants (as explained above in response to comment to L291), differences in $Phy_C$ are greater. The main reason for this discrepancy is the decoupling of $Phy_C$ and $Phy_N$ in the acclimative variants vs. their constant

proportionality as determined by the prescribed $Q$. As shown in Fig.7b, during winter and spring, $Q$ of the acclimative variants become much larger than the constant $Q$ of FS (computed as the biomass-weighed spatio-temporal averages of the $Q$ estimated by the IA variant, see Table 3). This, in turn, translates into much higher concentrations of $Phy_C$ as estimated by FS, compared to those estimated by the acclimative variants, despite the fact that the $Phy_N$ concentrations are similar among the model variants. Although this was hinted by the Fig.4d-f, a direct comparison of top 50m average concentrations, as shown in Fig. R1.3b below makes this clearer. The critical dependence of $Phy_C$ of the FS variant to the prescribed value of $Q$ was more clearly illustrated by a sensitivity experiment we conducted, where the FS variant was run with twice the default value of $Q$. This results in a dramatic reduction of simulated $Phy_C$ (Fig.R1.3d) without much difference in other quantities like DIN (not shown), $Phy_N$ and $\mu$ (Fig.R1.3e-f). In the revised manuscript, we will explain the differences in FS and acclimative variants along these lines, extend Fig.7 with $Phy_N$ and will mention the experiment with doubled constant $Q$, and possibly include the resulting equivalent of Fig.7 in an additional appendix.

[Figure]

Figure R1.3. Upper 50 m average (a,d) $Phy_C$ [mmolC m$^{-3}$], (b,e) $Phy_N$ [mmolN m$^{-3}$], and (c,f) $\mu$ [d$^{-1}$], as simulated by IA (fine-dashed dark blue line), DA (continuous orange line), (a-c) default FS (dashed green line) and (d-f) FS with doubled constant Q.

*"Discussion*
*L344-352 – the discussion in this paragraph mainly focuses in 3-D models, but this work considers a 1-D approach. For L349-350, the authors mention 'Recent applications of these models in 3D setups with realistic forcings (Kerimoglu et al., 2017; Pahlow et al., 2020) have indicated that accounting for acclimation enhances the ability of models to reproduce field observations.' I suggest to also giving examples of 1-D models that have been developed with flexible stoichiometry (photo-acclimation) and have been compared to simpler variants and have been a better fit for field observations."*
Response: Response: we will point to such examples (e.g., Christian, 2005; Ayata et al., 2013; Chen & Smith, 2018), including our own study that recently appeared (Anugerahanti et al., 2021, which was previously cited as 'in prep.'). Here we will also mention the utility of 1D setups in examining the behavior of model components in a cost-effective way (as was suggested in the next comment on the L365).

*"L365 – If computational costs were nearly negligible for the different variants in this work, adding a further state variable (e.g. zooplankton, phosphorus, etc) would have made the model more realistic and it would have been interesting to look at further biogeochemical processes in each of the variants and how they respond to the complexities added. I would suggest adding some thoughts in this paragraph about the importance of a 1-D model in terms of time- and space-effectiveness when referring to computational costs and to mention why not more state variables were added or if that is part of future work."*

Response: in this study, we chose a minimal setup for the sake of achieving a better understanding of the direct effects of the flexibilities involved in phytoplankton growth and nutrient uptake, and how these are modeled. As we explained below (in response to the comment on L450-454) in detail, inclusion of other food web components like zooplankton would make the model analysis substantially more difficult, and potentially inconclusive. We agree also that it is worth mentioning that computationally efficient 1-D setups are valuable environments by virtue of capturing the vertical structure of resource gradients, which has a determining role in functioning of the marine and aquatic systems. We will mention this in the context of earlier 1D studies regarding the phytoplankton acclimation (see our response to the previous comment on the L344-352). We had explained why phosphorus is not included in a paragraph in section 4.4, which we will reformulate this paragraph to make things more clear. Considering the comment below on the L416-419, we will include here the future plans of including P in the model in future work, and the relevance of this extension.

*"L387-388 'However, improvements in these specific aspects typically result in greater discrepancies in other aspects, such as the timing of the spring bloom, or winter concentrations of nutrients and phytoplankton.' - Was a different tuning of parameters tried? In Table 3, from the papers the authors used, how was it decided, which value to use? Was it a mean of all the estimates or different phytoplankton species? Could it have been a better tuning for the FS variant without facing higher discrepancies in the output? I suggest that in the caption of Table 3 it is specified about how that data was obtained from those two papers and mention in this section of the discussion if different parameterisations were tried and what issues were encountered. Currently, it sounds as if the higher discrepancies due to tuning are only an idea and not something proven."*

Response: In Table 3, the parameters were chosen as typical values from within the previously published range of values without particular reference to species, which we will clarify in the caption. We have extensively investigated the effects of parameters, but reporting of these results would be outside the scope of the current work. In fact, the observation stated in L387-388 was based on our work in tuning the FS and IA models against the observations in two oceanic sites, which we presented in a study that recently appeared (Anugerahanti et al., 2021), which was previously cited as `Anugerahanti et al' ('in prep' was omitted). Implications of particular choice of parameters, and investigating the behavior of models under different environmental conditions (which we have also partially explored, as hinted in L362-X) remain to be open questions, which we will mention explicitly.

*"L416-419 – Comparing to N:P ratios it is hard in this manuscript as the model does not include P. I suggest including more discussion about how the different variants would affect the N:P*

*ratio even if P is not a state variable in the model. This could be phrased as future work if the authors have interest in a follow-up for the N:P ratio."*

Response: We agree that discussing the N:P is difficult within the context of the present work, and while it is indeed the case that we are intending to address the N:P in a follow-up work, a discussion of this issue would be a source of distraction here. The statement here was regarding the qualitative role of acclimation in resource limitation. We will revise this sentence without particular reference to the N:P ratio, but will mention the plans on including P in a future study in a more fitting paragraph in section 4.4 (see our response above to the comment on L365).

*"L450-454 – Why was the NPZD model not used for this manuscript? Zooplankton is relevant for phytoplankton growth, especially in a location such as the one chosen in this manuscript, where seasonal changes are relevant. I think that there should be a strong justification as to why the NPZD model available from the FABM library was not used. How would the results be expected to change with an explicit zooplankton in each variant?"*

Response: As mentioned above, we chose a minimal setup in this study for being able to concentrate on the direct effects of the flexibilities involved in phytoplankton growth, and how these are described in models. Including zooplankton would have introduced indirect effects and complicate the model analysis. While being interesting and relevant in a broader context which we partially would like to address in follow-up work, having to deal with these intricacies would make it difficult to achieve our objectives of the current study. It is worth noting that the IA model can be comfortably coupled to a zooplankton module, based on our experience (Anugerahanti et al., in preparation). Coupling to zooplankton, in turn, would allow investigating how these bottom-up effects propagate through the food-web, and influence the ecosystem functions in a cost-effective way. We mentioned these in the extended discussion. We will mention these in this paragraph.

*"Overall, I think there should be a paragraph added with more discussion about the objectives set in this manuscript. Where the objectives fully addressed in this work?"*

Response: We believe that the stated objectives were addressed in this work, especially after revising the second objective following the suggestion of the reviewer, as explained above. The differences between the model variants are extensively discussed already in section 4.2. Therefore, we do not see the need for a repeated discussion on whether the objectives are met.

*"Conclusions*
*L472-479 – It is not necessary to mention again that information as it is clearly stated in the introduction and model description sections."*

Response:  we intended to provide here a wrap-up, but agree with the reviewer that it is too repetitive in its current form. Therefore we will condense this part.

*"For the conclusion, can the authors conclude more in terms of the objectives of the manuscript (section 1.4)? How do the authors justify the IA model without comparing it to observations and under an idealised set up? It is better or just different? If this is discussed in more detail in the*

*discussion section as suggested, then adding a couple of sentences about this in the conclusions would create a good closure for the manuscript."*

Response: As explained above, in the current study, we cannot argue that the IA model is better, but we show that it is substantially different, and as such, we will have fully met the second objective (in its revised form). However, we agree that a closure with reference to the specific objectives would indeed be good. We will revise the second paragraph accordingly.

*"Code availability – Please add the Zenodo link in this section."*

Response: we will add the Zenado link.

*"Technical corrections*
*Table 1 – in the Expansion/Value column, first row, do you mean Phyto$^2_N$ or Phyto$_N$?"*

Response: $Phy^2_N$ as typed is correct, as this is a quadratic mortality term (see the units of m in Table 1).

*"L127-132 – please be consistent with the quotation marks when you mention each model variant."*

Response: we will include the missing quotation marks.

*"Table 2 – What are the blank spaces representing?"*

Response: in fact there are no blank spaces: for instance Eq.(5) in the second row applies to all 3 variants. However this is indeed not clear when the vertical lines are not shown. In the revised manuscript, we will include the vertical lines if allowed by the journal, otherwise we will write equations for each column.

*"Table 2 – For dimensionless variables, please write 'dimensionless' instead of a hyphen"*

Response: we considered this suggestion, however we are concerned that this will just lead to unnecessary crowding of the tables and figures. According to the International System of Units (SI), the recommended unit for dimensionless quantities is 'one', but we think this can be even more confusing than the hyphen. At the end, given the clarifications in the caption, we are convinced that there is no room for confusion anyway, therefore would like to keep the hyphen.

*"Table 2 – for the units with (\*\*), it is inconsistent to put gChl mol C-1. Table 2 caption states that they are in gChl gC-1."*

Response: gChl gC-1 is used only for presentation (eg in figures). But apparently this note in the caption leads to confusion, therefore we will remove it.

*"Table 2 – Why are there Equations being named in the middle of the columns? Such as Eq. 14, Eq. 18, Eq. 26, Eq. 11 and the last NA? If they correspond to both IA and DA, please specify it in each column. Otherwise, it is fairly confusing."*

Response: yes, the referred equations apply to IA and DA, but we understand the concern. If allowed by the journal formatting requirements, we will include vertical border lines, otherwise we will specify equations for each column.

*"L237 – The authors mention Eq. 23 and Eq. 17, is this correct? I think instead of Eq. 23, it should be Eq. 28. Please correct me if I am wrong."*
Response: you are right, the correct reference should be Eq.28. In fact, the correct reference for the equation needing I should be Eq. 22 and not Eq.17. We will correct these.

*"L243 – n2 = 17m. Missing the n (greek letter)."*
Response: we will include it.

*"L274 – Fig 4 h-i instead of Fig 3 h-i."*
Response: will do.

*"L282 – Fig 5 e-f, h-i instead of Fig 5 e-f, h-j"*
Response: will do.

*"L283 – Is it Fig 5 j-l instead of Fig 5 n-o? Fig 5 n-o is for light limitation."*
Response: yes, we should have referred to panels j-l

*"Figure 5 caption – for dimensionless variables/parameters, write 'dimensionless' instead of a hyphen."*
Response: as indicated in our response above to the comment for Table 2, we would like to keep the hyphen.

*"Figure 6 – subplot (4,3,7) – change (h) for (g)."*
Response: thank you, we will change it.

*"Appendix A, L501 – subscripts for growth and Q should be a multiplication (last term in Eq. A1)."*
Response: will correct this.

We would like to thank the reviewer again for their very careful reading and patiently helping us in eliminating technical errors.

**References:**

Anugerahanti, P., Kerimoglu, O., & Smith, S. L. (2021). Enhancing Ocean Biogeochemical Models With Phytoplankton Variable Composition. *Frontiers in Marine Science*, *8*, 675428. https://doi.org/10.3389/fmars.2021.675428

Ayata, S.-D., Lévy, M., Aumont, O., Sciandra, A., Sainte-Marie, J., Tagliabue, A., & Bernard, O. (2013). Phytoplankton growth formulation in marine ecosystem models: Should we take into account photo-acclimation and variable stoichiometry in oligotrophic areas? *Journal of Marine Systems*, *125*, 29–40. https://doi.org/10.1016/j.jmarsys.2012.12.010

Chen, B., & Smith, S. L. (2018). Optimality-based approach for computationally efficient modeling of phytoplankton growth, chlorophyll-to-carbon, and nitrogen-to-carbon ratios. *Ecological Modelling*, *385*, 197–212. https://doi.org/10.1016/j.ecolmodel.2018.08.001

Christian, J. R. (2005). Biogeochemical cycling in the oligotrophic ocean: Redfield and non-Redfield models. *Limnology and Oceanography*, *50*(2), 646–657. https://doi.org/10.4319/lo.2005.50.2.0646

Guidi, L., Jackson, G. A., Stemmann, L., Miquel, J. C., Picheral, M., & Gorsky, G. (2008). Relationship between particle size distribution and flux in the mesopelagic zone. *Deep Sea Research Part I: Oceanographic Research Papers*, *55*(10), 1364–1374. https://doi.org/10.1016/j.dsr.2008.05.014

Moore, J. K., Doney, S. C., Kleypas, J. A., Glover, D. M., & Fung, I. Y. (2002). An intermediate complexity marine ecosystem model for the global domain. *Deep Sea Research Part II: Topical Studies in Oceanography*, *49*(1–3), 403–462. https://doi.org/10.1016/S0967-0645(01)00108-4

Moore, J. K., Doney, S. C., & Lindsay, K. (2004). Upper ocean ecosystem dynamics and iron cycling in a global three-dimensional model. *Global Biogeochemical Cycles*, *18*(4), n/a-n/a. https://doi.org/10.1029/2004GB002220

---

## Author Comment (AC4)

**Response to Reviewer #2**

We thank the reviewer for their review and suggestions. Below, the original comments of the reviewer are quoted verbatim, and our responses are provided right after each comment.

*"General comments*
*The article presents the first spatially-resolved implementation of an "Instant Acclimation" approach to modelling plankton ecosystem. The authors found that the Instant Acclimation model performance was very close to that of a more computationally expensive Dynamic Acclimation version. Both of these models where markedly different from a physiologically less realistic Fixed Stoichiometry model. This is an important contribution that will allow biogeochemically and ecologically important stoichiometric variation to be efficiently included in computationally expensive global ecosystem and biogeochemistry models.*
*The objectives of the article where clearly communicated, and the approach accurately described. From my perspective the main weakness of the article was the 'Model Description' section, which I felt could be made a lot more coherent. As it stands, the article attempts to describe three versions of the general model in parallel. As the model contains a number of subsections, each subsection needs to be described in multiple different ways before we can move on to the next subcomponent. I found that this made it hard to understand how each version of the model works as an integrated whole. My recommendation would therefore be to first describe the Dynamic Acclimation model in full, before going on to describe how the Instant Acclimation and Fixed Stoichiometry models deviate from this. This makes more sense to me, as both of the latter models are effectively simplified versions of the former."*
Response: we can understand the concern of the reviewer. In fact, in the first draft of the manuscript, the model description was structured exactly as suggested by the reviewer: first an uninterrupted description of the IA model, then an explanation of the differences in the FS and DA variants. The main problem with that approach was that the physical gap (in the order of 2-3 pages) between the formulations and logic made it difficult to see and understand the difference between the variants, as we had realized based on a friendly review. Moreover, the presented description follows the model code more naturally, noting that we do not have 3 different modules, but a single one where the individual process descriptions are chosen based on the specified variant. We would also like to point out that a full description of the DA model can be found in Pahlow et al. (2013) and the IA model can be found in Smith et al., (2016) with minor differences (as explained in Appendix A2). For these reasons, we would prefer to keep the structure of the model description in its current form, but indicate more clearly that the full description of the DA and IA variant can be found in the aforementioned publications.

*"My other main comment is that I did not see the benefit of varying the way photoacclimation is handled in the three models. This mechanism is included in many 'fixed stoichiometry' models, so it is not a unique benefit of the two more sophisticated approaches. Given the not insubstantial level of complexity in the rest of this article, I wondered if it might not make more sense to fix this part of the model across the three cases, and focus on the more novel developments in the C:N ratios."*

Response: we again understand the concern of the reviewer, and agree that photoacclimative response adds an additional layer of complexity. Nevertheless, in this study we would like to provide an analysis of the presented IA model together with its photoacclimative response in its entirety, as it was used in another study (Anugerahanti et al., 2021, which was earlier cited as 'in prep.') and was planned to be extended further in follow-up studies (e.g., Anugerahanti et al., in prep.). It should be noted that the behavior of the instantaneous photoacclimation model that we used here had not been analyzed in a vertically resolved setup until our study, therefore we consider the results presented to be useful in this regard as well. For these reasons we would like to keep the photoacclimative response in the model. Despite the additional complexities imposed by the photoacclimative response, we believe that our analysis is fairly complete, but we are fully open to any suggestions that may help improve our analysis and cover/point to any potentially open issues.

*"Specific comments*
*Line 34: 'Models that account for variations in cellular composition are indeed more likely to provide more realistic estimates' - Suggest 'Models that account for variations in cellular composition are in principal more likely to provide more realistic estimates'"*
Response: we agree that the suggested reformulation is more accurate, since no examples are cited to support the statement until that point. We changed the statement as suggested.

*"Line 68: 'The key assumption is that growth and nutrient uptake are at all times strictly balanced [w.r.t. the internal C:N stoichiometry of the cell]'"*
Response: thank you, we will add the statement in brackets to be more specific.

*"Line 76: 'the inclusion of transport terms may lead to additional complications'. Please could you explain how/why this leads to extra complications?"*
Response: We will include the following explanation: In a spatially structured environment, transport of cells with a certain internal state to a zone where the typical (average) cell composition is different, can create a `resource storage' effect (e.g., Grover, 2009; Kerimoglu et al., 2012). A typical example of this is nutrient-replete cells (as represented by high N:C) at the deeper layers diffusing towards the surface layers across the thermocline where the cells are typically nutrient starved. In principal, this effect can be resolved only by explicitly tracing the constituents of the cell dynamically.

*"Line 96: 'the trivial flux terms'. I do not see how these terms are trivial?"*
Response:  here, we are referring to the flux terms listed in Table 1 that are trivial (i.e., all except $F_{DIN–PhyN}$ and $F_{DIC–PhyC}$, which are indeed non-trivial). We will revise the sentence to clarify this point.

*"Eqns: 1-3. I found this notation a bit confusing. I wonder if simple word equations might be the most straightforward here? (e.g. dPhyN/dt = phytoplankton uptake - linear mortality). Failing that, I think substituting in the terms from Table 1 would be a lot clearer."*

Response: The flux terms make the source and target pools immediately clear, therefore we find them to be useful. However, we understand that for those who are not used to this notation, these can be difficult to read, therefore will add 1-word descriptions of the terms in underbraces.

*"Line 153: "(equivalently, relative size of the chloroplast, following Pahlow and Oschlies (2013)), fC:". I found this hard to understand. Have we switched to an entirely new idea here (fV to fC)? If so wouldn't it be better to separate out, instead of adding it on parenthetically?"*
Response: This is not a new idea, however we realize that our statement in parentheses might be misleading, therefore we will reformulate this in a separate sentence.

*"Equation 30: Which state variables are actually transported? Presumably not C for the IA or FS models. It is noted in the Discussion that C biomass is not conserved - I expect due to issues with advecting C in IA model. This should perhaps be discussed in a bit more detail."*
Response: all state variables are transported. For the IA and FS variants, $Phy_C$ is not a state variable, but a diagnostic variable calculated based on $Phy_N$ and $Q$ (Fig.1), therefore it is indeed not transported. We will stress that the $Phy_C$ is a state variable for the DA variant. We would like to clarify also that the presented N-based version of the model is fully closed, i.e., mass conserving with regard to N. Only if we had resolved a second nutrient, e.g., C, additional nutrient flux terms would have been required to satisfy the conservation of this second element. This issue has been explained previously by Smith et al. (2016). We had tried explaining this in L455-460 in the discussion, but we understand that this explanation was not clear enough, therefore we will revise the paragraph.

*"Figure 4: The third row of panels (g-i) are cited out of order in the legend, which is slightly confusing."*
Response: in fact, there is no inconsistency here: first the concentrations (Phytoplankton N, C and Chl) are introduced, then the ratios (N:C, Chl:C) are introduced in the figure. Referring to the variables in displayed order would lead to repetitions.

*"Results section: I think it would be worth noting any differences in system level functional parameters such as overall primary production and C export."*
Response: thank you for this suggestion, we very much like this idea. The water-column integrated net primary production rate and C export rate as estimated by the model variants are shown in Fig. R2.1 below. We will include this figure in the revised manuscript.

[Figure]

Figure R2.1 (a) Integrated Net Primary Production (NPP) Rate [mmolC m$^{-2}$ d$^{-1}$] and (b) Carbon export rate [mmolC m$^{-2}$ d$^{-1}$] as simulated by the IA (fine-dashed dark blue line), DA (continuous orange line) and the FS (dashed green line) variants.

**References:**

Anugerahanti, P., Kerimoglu, O., & Smith, S. L. (2021). Enhancing Ocean Biogeochemical Models With Phytoplankton Variable Composition. *Frontiers in Marine Science*, *8*, 675428. https://doi.org/10.3389/fmars.2021.675428

Grover, J. P. (2009). Is Storage an Adaptation to Spatial Variation in Resource Availability? *The American Naturalist*, *173*(2), E44–E61. https://doi.org/10.1086/595751

Kerimoglu, O., Straile, D., & Peeters, F. (2012). Role of phytoplankton cell size on the competition for nutrients and light in incompletely mixed systems. *Journal of Theoretical Biology*, *300*, 330–343. https://doi.org/10.1016/j.jtbi.2012.01.044

Pahlow, M., Dietze, H., & Oschlies, A. (2013). Optimality-based model of phytoplankton growth and diazotrophy. *Marine Ecology Progress Series*, *489*, 1–16. https://doi.org/10.3354/meps10449

Smith, S. L., Pahlow, M., Merico, A., Acevedo-Trejos, E., Sasai, Y., Yoshikawa, C., Sasaoka, K., Fujiki, T., Matsumoto, K., & Honda, M. C. (2016). Flexible phytoplankton functional type (FlexPFT) model: Size-scaling of traits and optimal growth. *Journal of Plankton Research*, *38*(4), 977–992. https://doi.org/10.1093/plankt/fbv038

---

## Author Response (AR1)

**Response to Reviews**

Dear editor, dear reviewers,

We would like to thank both referees for their comments, which helped us to significantly improve the manuscript.

Here is a list of changes we made in the manuscript:

- Included two new figures
    - Fig.8: showing the vertical profiles of some key quantities.
    - Fig.10: showing the seasonal course of two ecosystem level parameters.
- Modified Fig.7 and B2:
    - Corrected averaging errors caused by ignoring vertical variations in layer heights.
    - Added two additional panels.
- Provided a fuller account of the differences between the behavior of the model variants both in the Results and Discussion sections.
- Clarified the objectives in the Introduction and accordingly the Abstract and Conclusions sections, emphasizing on the mechanisms governing the differences between the model variants.
- Tidied up the Discussion section.
- Corrected various typographic errors.

Our point-to-point responses, where necessary, quoting appended or revised text in blue fonts are provided below.

Best regards,
Onur Kerimoglu, on behalf of all authors

**Response to Review #1**

We thank the reviewer for their careful reading of the manuscript, and constructive suggestions. Below, the original comments of the reviewer are quoted verbatim, and our responses are provided right after each comment, with an indication of the action taken, where necessary, quoting the revised text in blue fonts.

*"General comments*
*The manuscript describes three variants (FS, IA, and DA) in a biogeochemical model (FABM-NflexPD) coupled with a hydrodynamical model (GOTM). It clearly describes the differences between a fixed stoichiometry (FS) to more complex variants considering a dynamic acclimation (DA) and an instantaneous acclimation (IA).*
*Based on an idealized set up, the response of each model variant to irradiance and temperature is studied. Results show that adding higher complexity to the model creates differences in the model output that cannot be negligible. It is an interesting manuscript related to the scope of the journal and model development.*
*Overall, this manuscript presents an interesting approach to understand the dynamics of adding a flexible stoichiometry. However, the authors pose two objectives for this manuscript that are not fully addressed, so it is suggested that more discussion is added to fully address the objectives specified. It is also necessary that the authors specify why they used an NPD model instead of the NPZD model available from the FABM library. Moreover, I think the manuscript is lacking a more extensive discussion (or conclusions) about the novelty of this work, which could be done by, for example, extending the discussion (or conclusions) to explain why this work is relevant, given the model development is interesting but there is not a comparison to observations. Under the idealized set up of the different model variants created in this work and without comparison to observations, the authors should clearly state the applicability of their work and why it is important to have an IA variant considered in further biogeochemical modeling research.*
*This manuscript is within the standards of excellence of the journal, but the authors should address the comments suggested here."*
Response: we revised the conclusion section and abstract to better communicate how the objectives are met and novelty of the work, and extended the discussion on the applicability of the model and reasons to consider acclimation in biogeochemical modeling research. Regarding not having used an NPZD scheme, but an NPD scheme: indirect effects of zooplankton make it difficult to understand the direct effects of including acclimative mechanisms. Please see our detailed responses below to the specific comments.

*"Specific comments*
*Introduction*
*L29 It is mentioned that acclimation models are now commonly used, but only Geider's model is included. Could it be added a couple more examples apart from Geider's paper?"*
Response: ? We mentioned additional examples. We also realized that the example given for constant N:C and variable Chl:C ratio should not be Moore et al., (2002), where N:C was variable, but Moore et al. (2004), we corrected this. Now the revised part reads (L33-40):

'.. For describing variable Chl:C ratios, acclimation models, most commonly that of Geider et al. (1998), but also others (e.g. Pahlow and Oschlies, 2009; Wirtz and Kerimoglu, 2016) are being increasingly employed in biogeochemical model frameworks. Such models typically couple a description of variable N:C (or other nutrient:C) with photoacclimation, i.e., variation of Chl:C, using one more state variable for Chl bound to phytoplankton (Moore et al., 2002; Schourup-Kristensen et al., 2014; Kerimoglu et al., 2017; Kwiatkowski et al., 2018). Some models assume a constant N:C ratio, while describing the variations in Chl:C, e.g., using only the photoacclimation portion (e.g., Moore et al., 2004) of the model by Geider et al. (1998) or using an empirical function (e.g., Oschlies and Schartau, 2005), that was earlier proposed by Cloern et al. (1995).'

*"L34-35 In the sentence 'Models that account for variations in cellular composition are indeed more likely to provide more realistic estimates of phytoplankton biomass and biogeochemical fluxes'. Can it be briefly explained how are estimates provided more realistic based on the variable cellular ratios?"*
Response: we briefly explained how the estimates of the models are improved when the variable Chl:C and C:nutrient ratios are taken into account. This section now reads (L41-46)
'Models that account for variations in cellular composition are in principle more likely to provide more realistic estimates of phytoplankton biomass and biogeochemical fluxes: when the variabilities in Chl:C and C:nutrient ratios are realistically represented by the models, their calibration on the basis of in situ and satellite Chl observations become more accurate (Behrenfeld et al., 2009; Ayata et al., 2013; Kerimoglu et al., 2017), and their estimates of biosynthesis rates of C and nutrients, consequently the drawdown of nutrients, and elemental composition of the export flux can be better reproduced (Anderson and Pondaven, 2003; Mongin et al., 2003) respectively.'

*"L38 clones/types – what is the meaning of this? Could it be specified what are clones/types?"*
Response: we specified that these are 'phytoplankton functional types or clones' (L49).

*"L42 ... 'response to changes in resource environment' ... - please describe what those resources are."*
Response: we specified the resources as mineral nutrients and light (L54).

*"L48-54 Please add the units of Q, fv, fA, and the Chl:C."*
Response: we added units for Q as [molN/molC] and θ as [gChl/molC] and specified that fv, fA are dimensionless (L61-66).

*"L59-60 What is short-term in this case? Please specify."*
Response: we specified that here short-term refers to hours to days (L71).

*"L84-85 'Compared to the FS variant, do the results differ sufficiently to justify the additional complexities introduced by the IA variant?' - I think this is an interesting question to set as an objective of this study. However, it is not fully addressed in this work (see further comments in the conclusion section). If the FS model provides significantly different results than the IA model, is that a justification to add further complexity to models? Under an idealised setup and with no*

*comparison to observations, how can you tell that the differences found are answering to your question? Because of the obvious differences between FS and IA stated in section 1.4, it is expect that the results of the simulation will differ between both approaches. I suggest this question to be rephrased, as it is hard to justify IA over FS when there's no comparison to observations, which would help the manuscript better ground the differences found. Also, add more about the answer to your objective (ii) in the discussion and conclusion section (see below comments for those sections)."*

Response: thanks for this careful observation. Although earlier work have provided ample evidence for the improved realism of variable cellular composition models as explained in the previous paragraphs of our introduction, without any comparison against observation data, we indeed cannot defend in the current study that the performance of the IA variant is better than the FS variant, which the objective (ii) itself implies presumptively by '.. to justify the additional complexities introduced by the IA variant'. We simplified this objective as (L102):
'compared to the FS variant, do the results of the IA variant differ substantially?'.
We also added that (L102-103)
'While answering these questions, we aimed gaining mechanistic understanding of the dynamics driving the difference between the model estimates'.
Accordingly, we extended the relevant explanations in Results and Discussion sections (please see below for our response to the specific comments), and mentioned this aspect in the abstract (L15-17):
'Our analysis provides insights into the roles of acclimative flexibilities in simulated primary production and nutrient drawdown rates, seasonal and vertical distribution of phytoplankton biomass, formation of thin chlorophyll layers and stoichiometry of detrital material.'

*"Model description*
*L128 Which mechanisms mentioned in the Introduction is this referring to? I am not sure to what section of the introduction to look. Could it be specified which section? Or mention those mechanisms in L128."*

Response:  We are referring to the section 1.2 in the introduction. In the revised manuscript, we now referred to this section and mentioned these mechanisms explicitly as follows (L148-150):
'... acclimation mechanisms introduced in Section 1.2, here as represented by flexibilities in growth vs. nutrient uptake; nutrient affinity vs. maximum uptake; and chlorophyll density in chloroplasts; each of which are explained in detail in the following sections.'

*"L142, eq. 10 – Please describe what Qo is and reference where it is defined (Table 3)."*
Response: we described Qo and other intermediate terms appearing in the equation, referring to tables where they are defined (L165-166).

*"L177 – Please describe what nutrient affinity is in this section."*
 Response: We introduced the definitions for nutrient affinity, and what affinity and maximum nutrient uptake represent (L200-205):
"Originally introduced for describing the substrate uptake by bacteria, 'affinity' of a microorganism  `can be viewed as a measure of effective collusion between substrate and transport site' (Button, 1978), which can be practically found from the initial slope (i.e., before saturation) of the uptake rate with respect to the substrate concentration (Button, 1978). The term has been used for describing the nutrient uptake by phytoplankton (Aksnes and Egge,

1991), and recognized to be a measure of competitive ability under low concentrations. The maximum nutrient uptake rate, to the contrary, can taken to be a measure of competitiveness under high nutrient concentrations."

*"L203 – Add reference at the end of the sentence (Table 3)."*
Response: done (L232).

*"L248 – The sinking rate value was based on observations? Please say where was that value obtained from."*
Response: we indicated here (L277-279) that:
'.. value was arbitrarily chosen to induce a downward flux in this idealized setup, and that in reality, it depends on the average size and density of detritus particles being modelled and displays a vast range (Guidi et al., 2008).'

*"L254 – Mention the other meteorological variables assumed as constant."*
Response: we mentioned (L285) that the constant meteorological variables are wind speed, air pressure, humidity, cloud cover and precipitation (none).

*"Results*
*L287 – can the authors extend the explanation about the effect of DIN depletion? Are the DIN depletion differences the only reason for the differences between Ln (FS) and fc (IA + DA)?"*
Response: in the revised paragraph, we now referred to the exact DIN concentrations in a following paragraph (L336-338, please see our response and the quoted paragraph to the comment on L304-304). Differences between the $L_N$ (FS) and $f_C$ of the acclimative variants cannot be reduced to the differences in DIN because of the structural differences between how these quantities are computed, although they both serve in representing nutrient limitation of growth, thus they are functionally equivalent. But the incomplete depletion of DIN as simulated by the FS variant, and the high sensitivity of the Monod function (Eq.15) to the nutrients at low concentrations are the reasons for the $L_N$ values being not too low. This contrasts with the $f_C$ of IA and DA being close to zero, owed to $f_V$ being close to its maximum value of 0.5 and $Q$ being close to the subsistence quota, $Q_0$ (see Eq.11). We provided an explanation accordingly, however, later in a later paragraph (L344-351, quoted below in our response to the comment on L312-313).

*"L291 – In the contour plot for the net growth rate, it looks as if FS has higher values during the spring bloom in comparison to IA and DA."*
Response: thanks for noticing this, our statement does not apply for growth rate, but for nutrient uptake rate. We now explicitly pointed out to the higher growth rate as estimated by the FS variant during winter/spring (L323-324). FS attaining higher growth rates during the spring bloom indeed deserves some explanation, which is now provided as follows:
(L323-332) 'During winter and spring blooms, the net cellular growth rate, $\mu$ as estimated by the FS variant temporarily exceeds those estimated by the acclimative variants (Fig. 6a-c, see below for the explanation). The IA and DA variants estimate higher nutrient uptake rates, $V$, in surface layers during the spring bloom, and in deeper layers during summer (Fig. 6d-f).

Negative $V$ in the bottom layers as estimated by the FS and IA variants is a direct result of the balanced growth assumption (Eq. (6)) and can be interpreted as exudation. Respiratory costs of nutrient uptake, $R_N$, (Fig. 6h-i) are much lower than $R_{Chl}$ (Fig. 6j-l). For the FS variant, $R_N$ drops below 0 in the deeper (>50m) waters, implying negative respiration, which is a model artefact, as a result of $\hat{\mu}_{net}$ becoming negative (see Eq. (A4) in Sect.A1) due to the fixed $\hat{\theta}$. However these negative values are small, and therefore do not have a significant effect on the model results, as evidenced by a sensitivity experiment, where $\hat{\mu}_{net}$ was constrained to positive values for the FS variant (results not shown). In comparison to the acclimative variants, $R_{Chl}$ of the FS variant is smaller during the spring bloom, but larger during summer, reasons for which are explained below.'

(L352-364) 'The cellular net growth rate, $\mu$, as estimated by the FS variant is slightly faster than those of the acclimative variants during winter/spring near the surface (e.g. Fig. 8Ff,Mf) but becomes slower right after the spring bloom (e.g. Fig. 8Af), and stays low throughout the summer (Fig. 7f, Fig. 8Jf). It should be noted that, the chloroplast-specific growth rate, $\hat{\mu}$, which is maximized for the acclimative variants through photoacclimative flexibility (Section 2.2.4), is always higher than that calculated by the FS variant, as expected (not shown). As the chloroplast specific chlorophyll maintenance and synthesis costs, $\hat{R}_{Chl}$ is scaled to the cellular level (through multiplication with $f_C$, Eq. (25)), the resulting $R_{Chl}$ for the FS becomes lower than those of the acclimative variants, given that the prescribed $f_C$ of the FS variant during this time period is smaller than the dynamically calculated values by the acclimative variants (Fig. 7c; Fig. 8Fc,Mc). The lower $R_{Chl}$ of the FS variant, in turn, explains the higher $\mu$ during the spring bloom (Fig. 7f). When the chloroplast size of the FS variant is assumed to be proportional to $L_N$ as explained in the Appendix B, estimated growth rate becomes similar to those of the acclimative variants (Fig. B2f). During summer, this effect becomes reversed: high $R_{Chl}$ as estimated by the FS variant in the surface layers (Fig. 6j vs k-l) contributes to the relatively low $\hat{\mu}$ estimated by this variant (Fig. 7; Fig. 8Jf): in addition to the higher ^, the IA and DA variants achieve lower $R_{Chl}$ (Fig. 6j-l) through lower $\hat{\theta}$ (Fig. 5a-c) and fC (Fig. 7c, Fig. 8Jc) at the surface.'

To support these explanations, we now included the prescribed value of $f_C$ for the FS variant in Fig.7c and B2c and the additional panels (f) in both figures that show the top 50m average of $\mu$ (as were provisionally shown in Fig.R1.1 in the original responses we posted in the discussion forum). Please note also that, earlier, we had overseen the vertically heterogeneous layer thicknesses for the calculation of vertical averages in Figs 7, B2 and R1.1. This error is now corrected in the revised manuscript, which resulted in only minor deviations from the initial version.

*"L300 – Which sensitivity experiment?"*
Response: these results were not shown, we indicated this at the end of the sentence (L331).

*"L304-305 – saying 'before the onset of winter mixing' is vague. Do the authors mean one day before, the week before? I suggest it is specified which day/date this is. Moreover, it is confusing to see how 'substantially' higher are the values of DIN for the FS variant in comparison to IA and DA variants. To avoid confusion, please also specify what the DIN values are for the FS, IA, and DA variants before (insert date here) the onset of winter mixing. I can see*

*there are differences during the summer period for DIN, but is that a substantial difference?*
*Beware of the adjectives used if there is not a quantitative comparison."*
Response: we now mentioned 'early November' for the onset of winter mixing, and specified the exact DIN concentrations estimated by the FS and acclimative variants on the date the minimum concentrations are reached. The revised section (L337-339) is as follows:
'In both the IA and DA variants, DIN concentrations are almost entirely depleted before the onset of winter mixing in early November, with minimum concentrations of ~0.005 mmolN m$^{-3}$ near the surface. In the FS variant DIN remains higher (minimum concentration of ~0.7 mmolN m$^{-3}$ near the surface (Fig. 7a, Fig. 8Ja,Na).'
Relevantly, recognizing that referencing both times of the year and quantities in the contour plots are vague, we included an additional figure (new Fig.8. Note that this figure now includes the nutrient uptake rate, $V$, in addition to the parameters shown in the provisional Fig. R1.2 in our original response posted in the forum) that shows the vertical profiles at different times of the year.

*"L312-313 – the differences between Ln (FS) and fc (IA, DA) are large to be only explained by*
*small differences in the DIN during the summer period. Could the authors give a thought to that*
*difference and explain it in a short sentence?"*
Response: Differences in DIN at the near-surface are stronger (e.g., minimum concentrations reached by FS and IA/DA variants are 0.005 vs 0.7 mmolN m$^{-3}$, respectively, as explained now in the revised paragraph in L337-339, quoted above) than suggested by the top 50m averages shown in Fig.7c. Besides, as explained above in response to the comment on L287, the Monod function is strongly sensitive to the DIN at low concentrations. In the revised manuscript, we provided these extended explanations. We also realized that the last sentences in this paragraph can be potentially confusing therefore we removed them. Now the revised section reads (L344-351):
'During winter and the spring bloom in March-April, nutrient limitation is almost non-existent for the acclimative variants as indicated by $f_C$ approaching unity (Fig. 7c), whereas for the FS variant, a degree of nutrient limitation persists (as indicated by $L_N < f_C$), owed to the saturating behavior of the Monod function to the nutrient concentrations. During late summer (July to October), nutrient limitation becomes less severe for the FS variant than for the acclimative variants in the surface layers (i.e., $L_N > f_C$, Fig. 8Jc,Oc). The relatively high $L_N$ (minimum: 0.12) of the FS variant results from the incomplete DIN depletion as simulated by the FS variant as mentioned above, and the linear response of the Monod function to substrate concentrations at low levels (Eq. (15). In contrast, for the IA and DA variants, $Q$ approaches $Q_0$ and $f_V$ approaches its maximum value of 0.5, causing (through Eq. (11)) severe nutrient limitation, as $f_C$ approaches to zero (minimum: 0.005) near the surface.'

*"L316 – Please describe more the differences between FS and IA, DA during the spring bloom.*
*FS reaches a higher PhyC in March than IA and DA."*
Response: We agree that the differences between the model variants need to be better explained, and revised the text as follows:
(L365-375) 'During the spring bloom, C bound to phytoplankton, $Phy_C$, simulated by the FS variant exceeds those of the IA and DA variants (Fig. 7d), whereas the differences between the

N bound to phytoplankton, $Phy_N$ as simulated by different variants are much smaller (Fig. 7e). This discrepancy between C and N content of phytoplankton is due to the decoupling in the acclimative variants: due to the lower value of the prescribed $Q$ of the FS variant (based on the spatio-temporal average of the values simulated by IA) during winter-spring season (Fig. 7b), a larger amount of C-biomass can be synthesized per N taken up in comparison to the acclimative variants, explaining therefore the higher $Phy_C$ simulated by the FS. The sensitivity of $Phy_C$ of the FS variant is evidenced also by a strong reduction of $Phy_C$ (in contrast to relatively unaltered $Phy_N$) during the spring bloom in response to a doubling of the prescribed $Q$ (not shown). During summer, the FS variant estimates considerably lower values of $Phy_C$ compared to the IA and DA variants (Fig. 7d) whereas the simulated PhyN concentrations remain to be similar (Fig. 7e). Therefore the higher $Phy_C$ concentrations simulated by the acclimative variants during this period are promoted by lower $Q$ (Fig. 7b, Fig 8Jb) in the surface layers.

(L376-384) 'Differences between the IA and DA variants emerge especially right after the spring bloom and autumn destratification. After the spring bloom, growth rate simulated by the IA variant (until May) becomes lower than that by the DA variant (Fig. 7f) near the surface (Fig. 8Af). The main reason for this difference is the slightly lower $f_C$ of the IA variant during the winter-spring period (i.e., from December to May) near the surface (Fig. 8Fc,Mc,Ac) except for a short period at the peak of the bloom (Fig. 7c). The lower $f_C$ of IA during this period is, in turn, driven by slightly lower $Q$ (Fig. 7b, Fig. 8Mb,Ab), which also leads to slightly higher $f_V$ (see Eq. (14)). As pointed out above, the higher $Q$ simulated by the DA variant before the spring bloom near the surface is maintained by the homogenizing effect of vertical transport (which does not occur with the IA variant), and after the spring bloom following the onset of stratification, the persistently higher $Q$ of the DA variant near the surface is reflects the lagged response captured by dynamically tracing C and N content of phytoplankton.'

(L385-398) 'Following the weakening of stratification in early November (Fig. 3), a new phyto-plankton bloom develops, especially as reflected by $Phy_N$ in all variants, but also by $Phy_C$ as simulated by the DA variant (Fig. 7d,e). This bloom is driven by the entrainment of DIN and phytoplankton biomass below the thermocline into the SML (compare Fig. 8Oa,d,e vs. Fig. 8Na,d,e). Under these nutrient-replenished conditions, $\mu$ is predominantly limited by light, as in winter (Fig. 8Ff), therefore monotonically increases towards the surface (Fig. 8Nf), as simulated by all variants. On the other hand, vertical distribution of $Q$ as simulated by the IA and DA variants become qualitatively different: due to the rapid turbulent mixing of $Phy_C$ and $Phy_N$ as simulated by the DA variant, $Q$ is homogeneously distributed within the SML (Fig. 8Nb), but such homogenization does not occur in the IA variant, and $Q$ is determined by the locally optimized $f_V$. Therefore, in the DA variant, a high nutrient uptake at the bottom of the SML (Fig. 8Ng), in combination with mixing within the SML can support growth near the surface (through $Q$, Fig. 8Nb), whereas in IA, growth and uptake dynamics are always coupled by definition, and determined by local physiological states only, as in the FS variant. The decoupling of (growth and uptake) rates and re-shuffling of $Q$ as simulated by the DA variant appears to allow faster uptake of nutrients in comparison to the IA variant within the SML (Fig. 8Ng). A related mechanism potentially contributing to the higher nutrient uptake rates is again a time-lag effect: in the DA variant, the nutrient-starved phytoplankton (i.e., the low $Q$, see Fig. 8Ob) in the SML corresponds to a higher nutrient demand.

To support the explanations, we also extended Figs. 7 and B2 with $Phy_N$ (in both, panels e) and mentioned the experiment with doubled $Q$ only verbally without including an additional figure (i.e., the provisional R1.3 in our original response posted in forum), in order to avoid a further expansion of the already voluminous manuscript.

*"Discussion*
*L344-352 – the discussion in this paragraph mainly focuses in 3-D models, but this work considers a 1-D approach. For L349-350, the authors mention 'Recent applications of these models in 3D setups with realistic forcings (Kerimoglu et al., 2017; Pahlow et al., 2020) have indicated that accounting for acclimation enhances the ability of models to reproduce field observations.' I suggest to also giving examples of 1-D models that have been developed with flexible stoichiometry (photo-acclimation) and have been compared to simpler variants and have been a better fit for field observations."*
Response: we now pointed to such examples. We also mentioned the utility of 1D setups in examining the behavior of model components in a cost-effective way (as was suggested in the next comment on the L365). The revised section is as follows (L439-451):
'Earlier studies had pointed out that representation of variable in Chl:C:N ratios of phytoplankton in models resulted in better reproduction of field observations (e.g. Doney et al., 1996; Christian, 2005; Ayata et al., 2013; Chen and Smith, 2018). Consistent with those studies, implementation of the model introduced here for simulating two oligotrophic ocean sites suggested that the the portability of phytoplankton growth models are enhanced by the variable cellular composition Anugerahanti et al., 2021). As demonstrated by these studies, 1D setups, as we also used here are ideal computational environments for examining the behavior of phytoplankton growth models: while resolving the essential features of aquatic environments, foremost the seasonally variable vertical structuring of resources and transport rates, they increase the computational costs minimally, in comparison to the 3D models. On the other hand, realistic representation of the horizontal gradients, or investigation of the effects of phytoplankton on the biogeochemical functioning at larger scales do require 3D setups.'

*"L365 – If computational costs were nearly negligible for the different variants in this work, adding a further state variable (e.g. zooplankton, phosphorus, etc) would have made the model more realistic and it would have been interesting to look at further biogeochemical processes in each of the variants and how they respond to the complexities added. I would suggest adding some thoughts in this paragraph about the importance of a 1-D model in terms of time- and space-effectiveness when referring to computational costs and to mention why not more state variables were added or if that is part of future work."*
Response: in this study, we chose a minimal setup for the sake of achieving a better under-standing of the direct effects of the flexibilities involved in phytoplankton growth and nutrient uptake, and how these are modeled. As we explained below (in response to the comment on L450-454) in detail, inclusion of other food web components like zooplankton would make the model analysis substantially more difficult, and potentially inconclusive. We agree also that it is worth mentioning that computationally efficient 1-D setups are valuable environments by virtue of capturing the vertical structure of resource gradients, which has a determining role in functioning of the marine and aquatic systems. We mentioned this in the context of earlier 1D

studies regarding the phytoplankton acclimation (see our response to the previous comment on the L344-352). We had explained why phosphorus is not included in a paragraph in section 4.4, which we will reformulate this paragraph to make things more clear. Considering the comment below on the L416-419, we included here the future plans of including P in the model in future work, and the relevance of this extension. The revised paragraph in Section 4.4 is as follows (L580-588):

'For simplicity, we have traced only N here fully (e.g., no explicit DIC, but only DIN, see Eq. (4)) and the model is therefore conservative with respect to N, but not with respect to C. When multiple nutrient elements in the dissolved inorganic material pool (e.g., C, N, and P) are resolved, maintaining mass balance becomes more complicated under the IA assumption (see Smith et al., 2016; Ward, 2017). FABM-implementation of a carbon-based version of the model that resolves the C and N cycles is being currently developed, which are we are planing to present in a separate study. The extended model will be able to resolve 585 C, N, P and micronutrient cycles based on a common mass-balance formalism, and therefore allow investigating the validity of assuming instantaneous optimization of C:N:P:micronutrient ratios under various environmental conditions (relevantly, see Bonachela et al., 2013). However, for various ecological applications, especially those resolving multiple phytoplankton types, tracing only one nutrient element, as in the current study may be sufficient and more convenient.'

*"L387-388 'However, improvements in these specific aspects typically result in greater discrepancies in other aspects, such as the timing of the spring bloom, or winter concentrations of nutrients and phytoplankton.' - Was a different tuning of parameters tried? In Table 3, from the papers the authors used, how was it decided, which value to use? Was it a mean of all the estimates or different phytoplankton species? Could it have been a better tuning for the FS variant without facing higher discrepancies in the output? I suggest that in the caption of Table 3 it is specified about how that data was obtained from those two papers and mention in this section of the discussion if different parameterisations were tried and what issues were encountered. Currently, it sounds as if the higher discrepancies due to tuning are only an idea and not something proven."*

Response: In Table 3, the parameters were chosen as typical values from within the previously published range of values without particular reference to species, which we clarified now in the caption. We have extensively investigated the effects of parameters, but reporting of these results would be outside the scope of the current work. In fact, the observation stated in L387-388 was based on our work in tuning the FS and IA models against the observations in two oceanic sites, which we presented in a study that recently appeared (Anugerahanti et al., 2021), which was previously cited as `Anugerahanti et al' ('in prep' was omitted). Implications of particular choice of parameters, and investigating the behavior of models under different environmental conditions (which we have also partially explored, as hinted in L361) remain to be open questions, which we now mentioned (L494-495):

'How acclimative flexibilities impact the sensitivity of models to parameter perturbations remains to be an open question.'

*"L416-419 – Comparing to N:P ratios it is hard in this manuscript as the model does not include P. I suggest including more discussion about how the different variants would affect the N:P*

*ratio even if P is not a state variable in the model. This could be phrased as future work if the authors have interest in a follow-up for the N:P ratio."*

Response: We agree that discussing the N:P is difficult within the context of the present work, and while it is indeed the case that we are intending to address the N:P in a follow-up work, a discussion of this issue would be a source of distraction here. The statement here was regarding the qualitative role of acclimation in resource limitation. We revised this sentence without particular reference to the N:P ratio (L533-536), but mentioned the plans on including P in a future study in a more fitting paragraph in section 4.4 (see our response above to the comment on L365).

*"L450-454 – Why was the NPZD model not used for this manuscript? Zooplankton is relevant for phytoplankton growth, especially in a location such as the one chosen in this manuscript, where seasonal changes are relevant. I think that there should be a strong justification as to why the NPZD model available from the FABM library was not used. How would the results be expected to change with an explicit zooplankton in each variant?"*

Response:  As mentioned above, we chose a minimal setup in this study for being able to concentrate on the direct effects of the flexibilities involved in phytoplankton growth, and how these are described in models. Including zooplankton would have introduced indirect effects and complicate the model analysis. While being interesting and relevant in a broader context which we partially would like to address in follow-up work, having to deal with these intricacies would make it difficult to achieve our objectives of the current study. It is worth noting that the IA model can be comfortably coupled to a zooplankton module, based on our experience (Anugerahanti et al., in preparation). Coupling to zooplankton, in turn, would allow investigating how these bottom-up effects propagate through the food-web, and influence the ecosystem functions in a cost-effective way. We revised this paragraph by adding the following L573-579):

'An explicit consideration of zooplankton can expected to introduce additional complexities: depending on how zooplankton C and N co-limitation is described, variabilities in phytoplankton stoichiometry may affect zooplankton growth (e.g., Mitra et al., 2007; Branco et al., 2018; Kerimoglu et al., 2018) and in turn, depending on the parameterization of zooplankton excretion and remineralization processes, subsequent phytoplankton blooms may occur. While it was our explicit aim to avoid such complicated indirect effects and focus on the direct effects of acclimation mechanisms on phytoplankton growth in this study, coupling the presented model to a larger ecosystem model including herbivores and their predators would allow investigating the propagation of these effects throughout the food web in a cost-effective manner.'

*"Overall, I think there should be a paragraph added with more discussion about the objectives set in this manuscript. Where the objectives fully addressed in this work?"*

Response: We believe that the stated objectives were addressed in this work, especially after revising the second objective following the suggestion of the reviewer, as explained above. The differences between the model variants are extensively discussed already in section 4.2. Therefore, we do not see the need for a repeated discussion on whether the objectives are met.

*"Conclusions*

*L472-479 – It is not necessary to mention again that information as it is clearly stated in the introduction and model description sections."*

Response:  we intended to provide here a wrap-up, but agree with the reviewer that it is too repetitive in its current form. Therefore we condensed this part as follows (L597-601):

'In this study, we present a FABM-implementation of the 'NflexPD' model, and the behavior of three variants it can emulate: a Fixed Stoichiometry (FS) variant that lacks any acclimative flexibility and explicitly tracks only N bound to phytoplankton; a Dynamic Acclimation (DA) variant that resolves various acclimative flexibilities by explicitly tracking the C and N in phyto-plankton; and the Instantaneous Acclimation (IA) variant that resolves the same flexibilities as the DA variant, but by tracking the N in phytoplankton as in the FS variant.'

*"For the conclusion, can the authors conclude more in terms of the objectives of the manuscript (section 1.4)? How do the authors justify the IA model without comparing it to observations and under an idealised set up? It is better or just different? If this is discussed in more detail in the discussion section as suggested, then adding a couple of sentences about this in the conclusions would create a good closure for the manuscript."*

Response:As explained above, in the current study, we cannot argue that the IA model is better, but we show that it is substantially different, and as such, we will have fully met the second objective (in its revised form). However, we agree that a closure with reference to the specific objectives would indeed be good. Accordingly, we revised the second paragraph as follows (L602-618):

'By applying the NflexPD model coupled to an idealized, 1D water column model, we aimed to understand: *i*) whether and how the behavior of the IA and DA variants differ; and *ii*) whether and how the behavior of the acclimative variants differ from the non-acclimative, fixed stoichiometry variant. With regard to the first of our objectives, we found that behavior of IA 605 is stable and in many respects very similar to that of DA, although differences arise during the spring and autumn transitions, owing to the lagged response and vertical transport of nutrient quotas in the DA variant. With this, our study provides proof-of-concept that the IA approach is applicable in spatially-explicit setups, and hints at conditions under which deviations from the fully explicit variant can be expected. With regard to the second objective, we found substantial differences between the behavior of the FS and acclimative variants: with the particular parameterization we show-cased here, the acclimative variants 610 estimated smaller spring blooms, but sustained growth during summer and stronger nitrogen depletion in the surface layers, as well as steeper chlorophyll layers at the thermocline; and unlike the FS variant, they can reproduce the variabilities in C:N of particulate matter. Moreover, a subset of quantities estimated by the FS variant, such as the phytoplankton biomass and NPP rates were found to be strongly sensitive to the prescribed parameters such as *Q*, which, in this study was derived as a spatio-temporal average from the IA variant, but is typically an adjustable parameter, implying thus a higher degree of freedom. These qualitative differences provide insight into the impact of acclimative flexibilities on model response, and their ecosystem-scale implications. The model implementation presented here tracks only N as dissolved nutrient, which restricts its utility in biogeochemical studies that require a complete representation of the cycling of multiple elements, but it can be readily used in various ecological contexts.'

*"Code availability – Please add the Zenodo link in this section."*

Response: we added the Zenodo link.

*"Technical corrections*
*Table 1 – in the Expansion/Value column, first row, do you mean Phyto$^2_N$ or Phyto$_N$?"*
Response: $Phy^2_N$ as typed is correct, as this is a quadratic mortality term (see the units of m in Table 1).

*"L127-132 – please be consistent with the quotation marks when you mention each model variant."*
Response: we included the missing quotation marks.

*"Table 2 – What are the blank spaces representing?"*
Response: in fact there are no blank spaces: for instance Eq.(5) in the second row applies to all 3 variants. In the revised manuscript, we have included the vertical lines for these columns.

*"Table 2 – For dimensionless variables, please write 'dimensionless' instead of a hyphen"*
Response: we considered this suggestion, however we are concerned that this will just lead to unnecessary crowding of the tables and figures. According to the International System of Units (SI), the recommended unit for dimensionless quantities is 'one', but we think this can be even more confusing than the hyphen. At the end, given the clarifications in the caption, we are convinced that there is no room for confusion anyway, therefore would like to keep the hyphen.

*"Table 2 – for the units with (\*\*), it is inconsistent to put gChl mol C-1. Table 2 caption states that they are in gChl gC-1."*
Response: gChl gC-1 is used only for presentation (eg in figures). But apparently this note in the caption leads to confusion, therefore we removed it.

*"Table 2 – Why are there Equations being named in the middle of the columns? Such as Eq. 14, Eq. 18, Eq. 26, Eq. 11 and the last NA? If they correspond to both IA and DA, please specify it in each column. Otherwise, it is fairly confusing."*
Response: yes, the referred equations apply to IA and DA, but we understand the concern. As mentioned above, we included vertical lines in the Table, hoping everything will be clear now.

*"L237 – The authors mention Eq. 23 and Eq. 17, is this correct? I think instead of Eq. 23, it should be Eq. 28. Please correct me if I am wrong."*
Response: you are right, the correct reference should be Eq.28. In fact, the correct reference for the equation needing I should be Eq. 22 and not Eq.17. We corrected these.

*"L243 – n2 = 17m. Missing the n (greek letter)."*
Response: we included it.

*"L274 – Fig 4 h-i instead of Fig 3 h-i."*
Response: done.

*"L282 – Fig 5 e-f, h-i instead of Fig 5 e-f, h-j"*
Response:  done.

*"L283 – Is it Fig 5 j-l instead of Fig 5 n-o? Fig 5 n-o is for light limitation."*
Response:  yes, we should have referred to panels j-l, now corrected.

*"Figure 5 caption – for dimensionless variables/parameters, write 'dimensionless' instead of a hyphen."*
Response: as explained above, we would like to keep the hyphen to avoid crowding the figures.

*"Figure 6 – subplot (4,3,7) – change (h) for (g)."*
Response: thank you, we changed it. Please also note that we corrected the titles of panels d-f as $V$ (which were erroneously titled as $V_N$) to make them consistent with the text.

*"Appendix A, L501 – subscripts for growth and Q should be a multiplication (last term in Eq. A1)."*
Response: corrected.

We would like to thank the reviewer again for these very careful observations and patiently helping us eliminate technical errors.

**References cited in responses:**

Anugerahanti, P., Kerimoglu, O., & Smith, S. L. (2021). Enhancing Ocean Biogeochemical Models With Phytoplankton Variable Composition. *Frontiers in Marine Science*, *8*, 675428. https://doi.org/10.3389/fmars.2021.675428

Moore, J. K., Doney, S. C., Kleypas, J. A., Glover, D. M., & Fung, I. Y. (2002). An intermediate complexity marine ecosystem model for the global domain. *Deep Sea Research Part II: Topical Studies in Oceanography*, *49*(1–3), 403–462. https://doi.org/10.1016/S0967-0645(01)00108-4

Moore, J. K., Doney, S. C., & Lindsay, K. (2004). Upper ocean ecosystem dynamics and iron cycling in a global three-dimensional model. *Global Biogeochemical Cycles*, *18, GB4028*, https://doi.org/10.1029/2004GB002220

**Response to Review #2**

We thank the reviewer for their review and suggestions. Below, the original comments of the reviewer are quoted verbatim, our responses are provided right after each comment, with an indication of the action taken, where necessary, quoting the revised text in blue fonts.

*"General comments*
*The article presents the first spatially-resolved implementation of an "Instant Acclimation" approach to modelling plankton ecosystem. The authors found that the Instant Acclimation model performance was very close to that of a more computationally expensive Dynamic Acclimation version. Both of these models where markedly different from a physiologically less realistic Fixed Stoichiometry model. This is an important contribution that will allow biogeochemically and ecologically important stoichiometric variation to be efficiently included in computationally expensive global ecosystem and biogeochemistry models.*
*The objectives of the article where clearly communicated, and the approach accurately described. From my perspective the main weakness of the article was the 'Model Description' section, which I felt could be made a lot more coherent. As it stands, the article attempts to describe three versions of the general model in parallel. As the model contains a number of subsections, each subsection needs to be described in multiple different ways before we can move on to the next subcomponent. I found that this made it hard to understand how each version of the model works as an integrated whole. My recommendation would therefore be to first describe the Dynamic Acclimation model in full, before going on to describe how the Instant Acclimation and Fixed Stoichiometry models deviate from this. This makes more sense to me, as both of the latter models are effectively simplified versions of the former."*

Response: we can understand the concern of the reviewer. In fact, in the first draft of the manuscript, the model description was structured exactly as suggested by the reviewer: first an uninterrupted description of the IA model, then an explanation of the differences in the FS and DA variants. The main problem with that approach was that the physical gap (in the order of 2-3 pages) between the formulations and logic made it difficult to see and understand the difference between the variants, as we had realized based on a friendly review. Moreover, the presented description follows the model code more naturally, noting that we do not have 3 different modules, but a single one where the individual process descriptions are chosen based on the specified variant. We would also like to point out that a full description of the DA model can be found in Pahlow et al. (2013) and the IA model can be found in Smith et al., (2016) with minor differences. For these reasons, we kept the structure of the model description in its current form, but indicated more clearly that the full description of the DA and IA variants can be found in the aforementioned publications:
At the end of the verbal description of the DA variant (L150):
'A full description of this variant (including diazotrophy) can be found in Pahlow et al. (2013).'
At the end of the verbal description of the IA variant (L153):
'A full description of this variant can be found in Smith et al. (2016).'

*"My other main comment is that I did not see the benefit of varying the way photoacclimation is handled in the three models. This mechanism is included in many 'fixed stoichiometry' models,*

*so it is not a unique benefit of the two more sophisticated approaches. Given the not insubstantial level of complexity in the rest of this article, I wondered if it might not make more sense to fix this part of the model across the three cases, and focus on the more novel developments in the C:N ratios."*

Response: we again understand the concern of the reviewer, and agree that photoacclimative response adds an additional layer of complexity. Nevertheless, in this study we would like to provide an analysis of the presented IA model together with its photoacclimative response in its entirety, as it is used in another study (Anugerahanti et al., 2021, which was earlier cited as 'in prep.') and is being extended further, to be published in follow-up studies (e.g., Anugerahanti et al., in prep.). It should be noted that the behavior of the instantaneous photoacclimation model that we used here had not been analyzed in a vertically resolved setup until our study, therefore we consider the results presented to be useful in this regard as well. For these reasons we would like to keep the photoacclimative response in the model. Despite the additional complexities imposed by the photoacclimative response, we believe that our analysis is fairly complete, but we are fully open to any suggestions that may help improve our analysis and cover/point to any potentially open issues.

*"Specific comments*
*Line 34: 'Models that account for variations in cellular composition are indeed more likely to provide more realistic estimates' - Suggest 'Models that account for variations in cellular composition are in principal more likely to provide more realistic estimates'"*

Response: we agree that the suggested reformulation is more accurate, since no examples are cited to support the statement until that point. We changed the statement as suggested. Relevantly, please also note that we extended a brief explanation of how the model estimates are improved specifically by taking the variabilities in cellular composition into account (L42-46): 'Models that account for variations in cellular composition are in principle more likely to provide more realistic estimates of phytoplankton biomass and biogeochemical fluxes: when the variabilities in Chl:C and C:nutrient ratios are realistically represented by the models, their calibration on the basis of in situ and satellite Chl observations become more accurate Behrenfeld et al., 2009; Ayata et al., 2013; Kerimoglu et al., 2017), and their estimates of biosynthesis rates of C and nutrients, consequently the drawdown of nutrients, and elemental composition of the export flux can be better reproduced (Anderson and Pondaven,2003; Mongin et al., 2003) respectively.'

*"Line 68: 'The key assumption is that growth and nutrient uptake are at all times strictly balanced [w.r.t. the internal C:N stoichiometry of the cell]'"*

Response: thank you, we added the statement in brackets to be more specific (now L81).

*"Line 76: 'the inclusion of transport terms may lead to additional complications'. Please could you explain how/why this leads to extra complications?"*
*Response: we included the following explanation (L 90-94):*
'In a spatially structured environment, transport of cells with a certain internal state to a zone where the typical (average) cellular composition differs, can result in a spatial storage advantage (Grover, 2009). A typical example of this is nutrient-replete cells (as represented by

*high N:C) at the deeper layers diffusing towards the Surface Mixed Layer (SML) across the thermocline where the cells are typically nutrient starved (e.g., Kerimoglu et al., 2012). In principal, this effect can be resolved only by explicitly tracing the constituents of the cell dynamically.'*

*"Line 96: 'the trivial flux terms'. I do not see how these terms are trivial?"*
Response: here, we are referring to the flux terms listed in Table 1 that are trivial (i.e., all except $F_{DIN-PhyN}$ and $F_{DIC-PhyC}$, which are indeed non-trivial). We revised the sentence to clarify this point (L114-116):
'The formal definition and exact formulation of the flux terms ($F_{FROM-TO}$) in Eqs. (1–4) that are trivial (i.e., all except $F_{DIN-PhyN}$ and $F_{DIC-PhyC}$) are provided in Table 1.'

*"Eqns: 1-3. I found this notation a bit confusing. I wonder if simple word equations might be the most straightforward here? (e.g. dPhyN/dt = phytoplankton uptake - linear mortality). Failing that, I think substituting in the terms from Table 1 would be a lot clearer."*
Response: The flux terms make the source and target pools immediately clear, therefore we find them to be useful. However, we understand that for those who are not used to this notation, these can be difficult to read, therefore we added the processes represented by each term in underbraces.

*"Line 153: "(equivalently, relative size of the chloroplast, following Pahlow and Oschlies (2013)), fC:". I found this hard to understand. Have we switched to an entirely new idea here (fV to fC)? If so wouldn't it be better to separate out, instead of adding it on parenthetically?"*
Response: To clarify, this is not a new idea, however we realize that our statement in parentheses is misleading, therefore we rephrased this in a separate sentence as follows (L175):
'… the trade-off is specified in terms of the fraction of cellular nitrogen reserves allocated to nitrogen uptake ($f_V$), which linearly increases $V$, and decreases $\mu_g$, through decreasing the resources available for carbon fixation, $f_C$, which is interpreted as the relative size of the chloroplast (Pahlow and Oschlies, 2013)'.

*"Equation 30: Which state variables are actually transported? Presumably not C for the IA or FS models. It is noted in the Discussion that C biomass is not conserved - I expect due to issues with advecting C in IA model. This should perhaps be discussed in a bit more detail."*
Response:  all state variables are transported. For the IA and FS variants, $Phy_C$ is not a state variable, but a diagnostic variable calculated based on $Phy_N$ and $Q$ (Fig.1), therefore it is indeed not transported. We now stressed that the $Phy_C$ is a state variable for the DA variant. After the equations (1-4) we added (L128):
'It should be noted that the $Phy_C$ is resolved as a state variable only by the DA variant (Eq. (1b)).'
We would like to clarify also that the presented N-based version of the model is fully closed, i.e., mass conserving with regard to N. Only if we had resolved a second nutrient, e.g., C, additional nutrient flux terms would have been required to satisfy the conservation of this second element. This issue has been explained previously by Smith et al. (2016). We had tried explaining this in

L455-460 in the discussion, but we understand that this explanation was not clear enough, therefore we revised the paragraph as follows (L580-588):

'For simplicity, we have traced only N here fully (e.g., no explicit DIC, but only DIN, see Eq. (4)) and the model is therefore conservative with respect to N, but not with respect to C. When multiple nutrient elements in the dissolved inorganic material pool (e.g., C, N, and P) are resolved, maintaining mass balance becomes more complicated under the IA assumption (see Smith et al., 2016; Ward, 2017). FABM-implementation of a carbon-based version of the model that resolves the C and N cycles is being currently developed, which are we are planing to present in a separate study. The extended model will be able to resolve C, N, P and micro-nutrient cycles based on a common mass-balance formalism, and therefore allow investigating the validity of assuming instantaneous optimization of C:N:P:micronutrient ratios under various environmental conditions (relevantly, see Bonachela et al., 2013). However, for various ecological applications, especially those resolving multiple phytoplankton types, tracing only one nutrient element, as in the current study may be sufficient and more convenient.'

*"Figure 4: The third row of panels (g-i) are cited out of order in the legend, which is slightly confusing."*
Response: in fact, there is no inconsistency here: first the concentrations (Phytoplankton N, C and Chl) are introduced, then the ratios (N:C, Chl:C) are introduced in the figure. Referring to the variables in displayed order would lead to repetitions.

*"Results section: I think it would be worth noting any differences in system level functional parameters such as overall primary production and C export."*
Response: thank you for this suggestion, this is a very good idea. The water-column integrated Net Primary Production rate and C export rate as estimated by the model variants were shown in Fig.R2.1 in our original response posted to the discussion forum. We realized in the meanwhile that the C export rates that we calculated at the bottom of the simulated water column are misleading, since we use no-flux boundary conditions at the bottom, such that the material accumulates in the bottom layers, amplifying therefore the estimated C-export rates. This issue could be fixed by calculating the fluxes not across the very bottom of the water column, but, at a given depth closer to the surface (e.g., the photic depth). However, considering that C-export rate is mainly a lagged and delayed NPP signal, we decided that nutrient drawdown (consumption) rate would be a more independent metric, which would also be more centrally related to the subject matter of our study. Therefore, in the new Fig. 10 we included in the revised version of the manuscript, we included the NDD rate, instead of the C export rate (we did not update the figure in the response letter to avoid potential confusions). We describe these results as follows (L404-418):

'Simulated process rates determining ecosystem functioning, such as the water-column integrated Net Primary Production (NPP) and Nutrient Drawdown (NDD) rates also differ between the model variants. FS estimates higher NPP rates during winter and the spring bloom (Fig. 10a), consistent with the higher $Phy_C$ it estimates during this period (Fig. 7d). While the NPP estimates of IA and DA are very close between the late summer (starting from September) to the spring bloom (in early March), right after the spring bloom, IA estimates suddenly decrease, as a consequence of reduced net specific growth rate, $\mu$ (Fig. 7f) as was pointed out

above. Interestingly, this difference between the IA and DA is larger than the differences in $\mu$, and contrasts with the differences in $Phy_C$ averaged 410 over the top 50m (Fig. 7d,f), but can be explained by the higher vertical covariance between the $Phy_C$ and $\mu$ in DA than in IA (Fig. 8Ad,Af). Annual average NPP rates as estimated by the FS (48.77 mmolC m$^{-2}$ d$^{-1}$) and IA (45.66 mmolC m$^{-2}$ d$^{-1}$) variants are respectively 8.1% and 13.9% smaller than that of the DA variant (53.06 mmolC m$^{-2}$ d$^{-1}$). NDD rates (Fig. 10b) are similar during the spring bloom, but the acclimative variants become higher during summer. After the autumn mixing, NDD as simulated by the DA variant shows a spike not well reproduced by the IA and FS variants, which is driven by the fast uptake rates simulated by the DA variant throughout the SML, contrasting with those simulated by the IA variant constrained to the surface layers (Fig. 8Ng). Annual average NDD rate simulated by the DA variant (4.78 mmolN m$^{-2}$ d$^{-1}$) is the highest, followed by the 8.2% lower IA (4.39 mmolN m$^{-2}$ d$^{-1}$) and 14.3% lower FS (4.1 mmolN m$^{-2}$ d$^{-1}$) variants.'

Later in section 4.2, we discuss these findings as follows (L513-525):

'We also found differences in system-level metrics such as NPP and NDD (e.g., Bergeron and Tremblay, 2014; Johnson et al., 2017) rates as simulated by different variants. For both metrics, DA estimates were about 10% higher than the FS and IA variants, with FS estimates system-atically skewed towards earlier in the season. It should be noted that, for the FS variant, prescribed $Q$, which, in this study was based on the results of the IA variant, but normally is effectively a free parameter (although the common approach is to set it to the Redfield pro-portions), largely determines the estimated $Phy_C$, and related quantities, such as NPP rates. For instance, doubling the $Q$ of FS results in only a few percent further underestimation (relative to DA) of the NDD rate (annual average: 3.89, instead of the original 4.1 mmolN m$^{-2}$ d$^{-1}$, which corresponds to 18.6% lower than the DA estimate, instead of the original 14.3%), whereas it leads to more than 50% lower estimates of NPP rate (23.26 mmolC m$^{-2}$ d$^{-1}$) in comparison to that of the DA variant. Some FS variants are not based on C, and not N as in this study (i.e., the explicit state variable is the C bound to phytoplankton). For those models, instead of the NPP, NDD rates may be more sensitive to prescribed $Q$. In contrast to the FS variant, with the IA variant, the total C and N content, and growth and nutrient uptake rates of the phytoplankton, thus the system-level process rates, like the PPR and NDD rates are determined by the same set of parameters governing the fully explicit DA variant.'

**References cited in responses:**

Anugerahanti, P., Kerimoglu, O., & Smith, S. L. (2021). Enhancing Ocean Biogeochemical Models With Phytoplankton Variable Composition. *Frontiers in Marine Science, 8*, 675428. https://doi.org/10.3389/fmars.2021.675428

Pahlow, M., Dietze, H., & Oschlies, A. (2013). Optimality-based model of phytoplankton growth and diazotrophy. *Mar. Ecol. Prog. Ser., 489*, 1–16. https://doi.org/10.3354/meps10449

Smith, S. L., Pahlow, M., Merico, A., Acevedo-Trejos, E., Sasai, Y., Yoshikawa, C., Sasaoka, K., Fujiki, T., Matsumoto, K., & Honda, M. C. (2016). Flexible phytoplankton functional type (FlexPFT) model: Size-scaling of traits and optimal growth. *Journal of Plankton Research, 38*(4), 977–992. https://doi.org/10.1093/plankt/fbv038